# Sensitization of colonic nociceptors by TNFα is dependent on TNFR1 expression and p38 MAPK activity

Katie H. Barker[1] , James P. Higham[1], Luke A. Pattison[1], Toni S. Taylor[1], Iain P. Chessell[2], Fraser Welsh[2], Ewan St. J. Smith[1] and David C. Bulmer[1]

[1]*Department of Pharmacology, University of Cambridge, Cambridge, UK*
[2]*Neuroscience, BioPharmaceuticals R&D, AstraZeneca, Cambridge, UK*

Handling Editors: Laura Bennet & Bernard Drumm

The peer review history is available in the Supporting information section of this article (https://doi.org/10.1113/JP283170#support-information-section).

**Abstract** Visceral pain is a leading cause of morbidity in gastrointestinal diseases, which is exacerbated by the gut-related side-effects of many analgesics. New treatments are needed and further understanding of the mediators and mechanisms underpinning visceral nociception in disease states is required to facilitate this. The pro-inflammatory cytokine TNFα is linked to pain in both patients with inflammatory bowel disease and irritable bowel syndrome, and has been shown to sensitize colonic sensory neurons. Somatic, TNFα-triggered thermal and mechanical hypersensitivity is mediated by TRPV1 signalling and p38 MAPK activity respectively,

This article was first published as a preprint: Barker K.H., Higham J.P., Pattison L.A., Taylor T.S., Chessell I.P., Welsh F., Smith E.S., Bulmer D.C. (2022). Sensitisation of colonic nociceptors by TNFα is dependent on TNFR1 expression and p38 MAPK activity. bioRxiv. https://doi.org/10.1101/2022.02.06.478183

downstream of TNFR1 receptor activation. We therefore hypothesized that TNFR1-evoked p38 MAPK activity may also be responsible for TNF$\alpha$ sensitization of colonic afferent responses to the TRPV1 agonist capsaicin, and noxious distension of the bowel. Using $Ca^{2+}$ imaging of dorsal root ganglion sensory neurons, we observed TNF$\alpha$-mediated increases in intracellular $[Ca^{2+}]$ and sensitization of capsaicin responses. The sensitizing effects of TNF$\alpha$ were dependent on TNFR1 expression and attenuated by p38 MAPK inhibition. Consistent with these findings, *ex vivo* colonic afferent fibre recordings demonstrated an enhanced response to noxious ramp distention of the bowel and bath application of capsaicin following TNF$\alpha$ pre-treatment. Responses were reversed by p38 MAPK inhibition and absent in tissue from TNFR1 knockout mice. Our findings demonstrate a contribution of TNFR1, p38 MAPK and TRPV1 to TNF$\alpha$-induced sensitization of colonic afferents, highlighting the potential utility of these drug targets for the treatment of visceral pain in gastro-intestinal disease.

(Received 21 May 2022; accepted after revision 28 June 2022; first published online 1 July 2022)

**Corresponding author** Dr David Bulmer, Department of Pharmacology, University of Cambridge, Tennis Court Road, Cambridge CB2 1PD, UK. Email: dcb53@cam.ac.uk

**Abstract figure legend** TNF$\alpha$ sensitized $Ca^{2+}$ responses to the TRPV1 agonist capsaicin in dorsal root ganglion sensory neurons. Sensitization was TNFR1-dependent and attenuated by inhibition of p38 MAPK. Direct $Ca^{2+}$ responses to TNF$\alpha$ were TRPV1 dependent. In *ex vivo* colonic afferent recordings, TNF$\alpha$ increased sensitivity to noxious ramp distension and capsaicin, both of which were absent in $TNFR1^{-/-}$ tissue or blocked by inhibition of p38 MAPK. These findings establish a role for TNFR1, p38 MAPK and TRPV1 in TNF$\alpha$-mediated sensitization of colonic afferents.

### Key points

- The pro-inflammatory cytokine TNF$\alpha$ is elevated in gastrointestinal disease and sensitizes colonic afferents via modulation of TRPA1 and $Na_V1.8$ activity. We further develop this understanding by demonstrating a role for p38 MAPK and TRPV1 in TNF$\alpha$-mediated colonic afferent sensitization. Specifically, we show that:
- TNF$\alpha$ sensitizes sensory neurons and colonic afferents to the TRPV1 agonist capsaicin.
- TNF$\alpha$-mediated sensitization of sensory neurons and colonic nociceptors is dependent on TNFR1 expression.
- TNF$\alpha$ sensitization of sensory neurons and colonic afferents to capsaicin and noxious ramp distension is abolished by inhibition of p38 MAPK.
- Collectively these data support the utility of targeting TNF$\alpha$, TNFR1 and their downstream signalling via p38 MAPK for the treatment of visceral pain in gastrointestinal disease.

## Introduction

Abdominal pain is a leading cause of morbidity in gastro-intestinal diseases, such as inflammatory bowel disease (IBD), and irritable bowel syndrome (IBS) (Grundy et al., 2019; Spiller & Major, 2016). Mechanistically, the sensitization of visceral nociceptors by inflammation, infection, allergy and other pathological processes, is a principal cause of pain in disease states, although the mediators and mechanisms underpinning this are not yet fully understood (Grundy et al., 2019;

**Katie Barker** graduated with a BSc in Pharmacology from Newcastle University, UK, and completed her MRes in Cancer Research at the Northern Institute of Cancer Research. She is currently completing her PhD at the University of Cambridge under the supervision of Dr David Bulmer and Professor Ewan St John Smith. Her research focusses on the role of pro-inflammatory cytokines in visceral pain, with a particular interest in gastrointestinal disease.

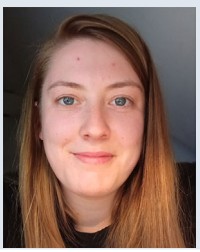

McMahon et al., 2015). The pro-inflammatory cytokine tumour necrosis factor $\alpha$ (TNF$\alpha$) has been implicated in the development of peripheral sensitization and visceral pain in IBD and IBS patients, based on its causative role in inflammatory disease pathology, localized release from mast cells, and the significant correlation between pain scores and peripheral blood mononuclear cell (PBMC)-evoked TNF$\alpha$ release in IBS patients (Hughes et al., 2013; Rijnierse et al., 2006). These observations are supported by expression of the TNFR1 receptor in mouse colonic sensory neurons (Hockley et al., 2019) and human dorsal root ganglion (DRG) neurons (Wangzhou et al., 2020), as well as functional data demonstrating the sensitization of colonic neurons and colonic afferents by TNF$\alpha$ (Hughes et al., 2013). These effects have been linked to enhanced tetrodotoxin (TTX)-resistant voltage-gated sodium channel currents (Na$_V$1.8 subtype) (Jin & Gereau IV, 2006) and suppressed voltage-gated potassium channel currents (delayed rectifier sub-types), downstream of TNFR1 receptor activation in DRG neurons (Ibeakanma & Vanner, 2010), as well as there being a role for TRPA1 channel activity in colonic afferents ((Hughes et al., 2013). The translational importance of these findings is supported by studies utilizing IBD patient colonic biopsy supernatants or IBS patient PBMCs, which have confirmed the essential contribution of TNF$\alpha$ to the respective sensitization of DRG neurons or colonic afferents using tissue from TNFR1$^{-/-}$ mice or pre-treatment with the anti-TNF$\alpha$ monoclonal antibody infliximab (Hughes et al., 2013; Ibeakanma & Vanner, 2010).

In addition to visceral pain, TNF$\alpha$ also evokes somatic, thermal and mechanical hypersensitivity by increasing TRPV1 activity (Khan et al., 2008) and p38 mitogen-activated protein kinase (MAPK)-mediated Na$_V$1.8 activity, downstream of TNFR1 receptor activation (Jin & Gereau IV, 2006). Given TRPV1 and Na$_V$1.8 channels are co-expressed with TNFR1 in colonic nociceptors (Hockley et al., 2019), we reasoned that TNF$\alpha$ may sensitize responses to the TRPV1 agonist capsaicin, and that p38 MAPK signalling may be responsible for TNF$\alpha$-mediated sensitization of colonic nociceptors. Consequently, the aims of this study were to investigate the contribution of p38 MAPK and TNFR1 to TNF$\alpha$-mediated sensitization of colonic afferents to capsaicin and noxious distension of the bowel using a combination of Ca$^{2+}$ imaging of DRG neurons and *ex vivo* electrophysiological recordings of colonic afferent activity.

## Materials and methods

### Ethical approval

All animal experiments were conducted in compliance with the Animals (Scientific Procedures) Act 1986 Amendment Regulations 2012 under Project Licence P7EBFC1B1 granted to E. St J. Smith by the Home Office and approval by the University of Cambridge Animal Welfare Ethical Review Body.

### Reagents

Stock concentrations of TNF$\alpha$ (0.1 mg/ml; H$_2$O with 0.2% (w/v) bovine serum albumin), capsaicin (1 mM; 100% ethanol) and staurosporine (10 mM; DMSO) were dissolved as described, all purchased from Sigma-Aldrich. R7050 (10 mM; DMSO), thapsigargin (1 mM; DMSO), A425619 (1 mM; DMSO) and SB203580 (10 mM; DMSO) were obtained from Tocris and stock concentrations made up as described. Nifedipine (100 mM; DMSO) and atropine (100 mM; 100% ethanol) were purchased from Sigma-Aldrich and dissolved as described. All drugs were diluted to working concentrations in extracellular solution (ECS) or Krebs buffer on the day of the experiment.

### Animals

Adult male C57BL/6J mice (8–16 weeks old) were obtained from Charles River (Cambs, UK; RRID:IMSR_JAX:000664). Mice were conventionally housed in temperature-controlled rooms (21°C) with a 12 h light/dark cycle and provided with nesting material, a red plastic shelter and access to food and water *ad libitum*. TNFR1$^{-/-}$ mice (9–14 weeks; 8F, 7M; Jackson Laboratory, ME, USA; RRID:IMSR_JAX:0 03242) were housed in individually ventilated plastic cages under the same conditions.

### Primary culture of mouse dorsal root ganglion neurons

DRG neurons were cultured as previously described (Hockley et al., 2019). Briefly, mice were killed by exposure to a rising concentration of CO$_2$, followed by cervical dislocation. Isolated DRG (T12-L5, spinal segments innervating the distal colon) were incubated in 1 mg/ml collagenase (15 min) followed by trypsin (1 mg/ml) both with 6 mg/ml BSA in Leibovitz's L-15 Medium, GlutaMAX Supplement (supplemented with 2.6% (v/v) NaHCO$_3$). DRG were resuspended in 2 ml Leibovitz's L-15 Medium, GlutaMAX Supplement containing 10% (v/v) fetal bovine serum (FBS), 2.6% (v/v) NaHCO$_3$, 1.5% (v/v) glucose and 300 units/ml penicillin and 0.3 mg/ml streptomycin (P/S). DRG were mechanically dissociated, centrifuged (100 *g*) and the supernatant collected for five triturations. Following centrifugation and resuspension, the supernatant (50 $\mu$L) was plated onto 35 mm poly-D-lysine-coated glass bottom culture dishes

(MatTek, MA, USA), and further coated with laminin (Thermo Fisher: 23017015). Dishes were incubated for 3 h to allow cell adhesion, before flooding with 2 ml Leibovitz's L-15 Medium, GlutaMAX Supplement containing 10% (v/v) FBS, 2.6% (v/v) NaHCO$_3$, 1.5% (v/v) glucose and P/S and cultured for 24 h. All incubations were carried out at 37°C with 5% CO$_2$.

## Ca$^{2+}$ imaging

Extracellular solution (in mM: 140 NaCl, 4 KCl, 1 MgCl$_2$, 2 CaCl$_2$, 4 glucose, 10 Hepes) was prepared and adjusted to pH 7.4 using NaOH and an osmolality of 290−310 mOsm using sucrose. Cells were incubated for 30 min with 100 $\mu$l of 10 $\mu$M Ca$^{2+}$ indicator Fluo-4-AM (room temperature; shielded from light). For inhibitor studies requiring pre-incubation, 200 $\mu$l of drug was added for 10 min prior to imaging.

Dishes were mounted on the stage of an inverted microscope (Nikon Eclipse TE-2000S) and cells were visualized at 10x magnification with brightfield illumination. Cells were initially superfused with ECS, or drug during inhibitor studies to establish baseline. For studies in which Ca$^{2+}$ was absent from the ECS, a bath solution was made up as follows (in mM): 140 NaCl, 4 KCl, 2 MgCl$_2$, 4 glucose, 10 Hepes, 1 EGTA (pH 7.35–7.45 with NaOH; 290−310 mOsm with sucrose). To compensate for the loss of extracellular divalent cations, the MgCl$_2$ concentration was increased and EGTA was used to chelate any remaining Ca$^{2+}$.

Fluorescent images were captured with a charge-coupled device camera (Rolera Thunder, Qimaging, MC, Canada or Retiga Electro, Photometrics, AZ, USA) at 2.5 fps with 100 ms exposure and a 470 nm light source for excitation of Fluo-4-AM (Cairn Research, Faversham, UK). Emission at 520 nm was recorded with $\mu$Manager (Edelstein et al., 2014). All protocols began with a 10 s baseline of ECS before drug super-fusion. With multiple drug additions to the same dish, cells were allowed 4 min recovery between applications. Finally, cells were stimulated with 50 mM KCl for 10 s to determine cell viability, identify neuronal cells and allow normalization of fluorescence. A fresh dish was used for each protocol and all solutions were diluted in ECS.

## Ca$^{2+}$ imaging data analysis

Individual cells were circled on a brightfield image and outlines overlaid onto fluorescent images using ImageJ (NIH, MA, USA). Pixel intensity was measured and analysed with custom-written scripts in RStudio (RStudio, MA, USA). Background fluorescence was subtracted from each cell, and fluorescence intensity (F) baseline corrected and normalized to the maximum fluorescence elicited during 50 mM KCl stimulation (F$_{pos}$). Maximum KCl fluorescence was denoted as 1 F/F$_{pos}$. Further analysis was confined to cells with a fluorescence increase $\geq$5 standard deviations above the mean baseline before 50 mM KCl application. Neurons were deemed responsive to a drug challenge if a fluorescence increase of 0.1 F/F$_{pos}$ was seen in response to drug perfusion. The proportion of responsive neurons and magnitude of the fluorescence response was measured for each experiment, with peak responses calculated from averaging fluorescence values of individual neurons at each time point.

## Magnetic-activated cell sorting

To determine the role of satellite cells, such as glia, in the neuronal responses to TNF$\alpha$, satellite cells were removed from DRG cultures using magnetic-activated cell sorting (MACS), with equipment purchased from Miltenyi Biotec and using protocols previously described (Thakur et al., 2014).

DRG from 2−3 mice were isolated and cultured as above, but trypsin incubation was omitted and DRG were incubated with collagenase (1 mg/ml with 6 mg/ml BSA) for 45 min. Pelleted neurons were washed in 2 ml Dulbecco's phosphate-buffered saline (DPBS, containing 0.9 mM CaCl$_2$ and 0.5 mM MgCl$_2$) and centrifuged for 7 min (100 $g$). The pellet was resuspended in MACS rinsing solution (120 $\mu$l), supplemented with 0.5% w/v BSA (sterile filtered at 0.2 $\mu$M), and incubated (5 min at 4°C) with a biotin-conjugated non-neuronal antibody cocktail (30 $\mu$l). DPBS was added to a volume of 2 ml and the suspension centrifuged for 7 min at 100 $g$. The pellet was resuspended in 120 $\mu$l MACS rinsing solution with 30 $\mu$L biotin-binding magnetic beads and incubated for a further 10 min at 4°C, before being topped up to 500 $\mu$l with MACS buffer.

The cell suspension was filtered by gravity through a magnetic column (LD column), primed with 2.5 ml MACS rinsing solution. Following the addition of the cell suspension, 1 ml MACS rinsing solution was used to collect the remnants of the cell suspension and passed through the column prior to a final wash. The 5 ml elute was centrifuged for 7 min at 100 $g$ and the final pellet resuspended in supplemented L-15 medium, before plating on 35 mm poly-D-lysine-coated glass bottom culture dishes further coated with Matrigel (diluted 1:10 in L-15 medium). Dishes were incubated for 3 h to allow cell adhesion, after which 2 ml Leibovitz's L-15 Medium, GlutaMAX Supplement containing 10% (v/v) FBS, 2.6% (v/v) NaHCO$_3$, 1.5% (v/v) glucose and P/S was added, and dishes were cultured for 48 h (37°C, 5% CO$_2$). Medium was changed after 24 h.

## Immunocytochemistry of cultured DRG neurons

DRG neurons were cultured as above and seeded onto 12 mm coverslips coated in poly-L-lysine and laminin. After 24−48 h in culture, cells were fixed at room temperature in 4%, pH 7.0 paraformaldehyde (10 min) and washed in PBS. Cells were permeabilized with 0.05% Triton-X100 for 5 min at room temperature. Cells were washed again in PBS and then blocking buffer (1% goat serum in 0.2% Triton-X100) was applied. After 2 h, cells were incubated with a rabbit anti-$\beta$III-tubulin primary antibody (1:1000, Abcam: ab18207; RRID:AB_444319) (Prado et al., 2021) for 3 h at room temperature.

Following primary antibody incubation, cells were washed in PBS and incubated with an Alexa Fluor-568 goat anti-rabbit secondary antibody diluted in PBS (1:1000, Invitrogen: A11008; RRID:AB_143165) (Crerar et al., 2019) plus 4'-6-diamidino-2-phenylindole (DAPI; 1:1000, Abcam) for 1 h at room temperature. After a final wash, coverslips were mounted, cell side down, on $25 \times 75 \times 1$ mm glass slides using Mowoil 4−88 mounting medium (Sigma-Aldrich: 81381). Mounting medium was set at 4°C and slides were imaged within 1 h.

Slides were imaged using an Olympus BX51 microscope. Fluorophores were excited with 568 nm (Alexa Fluor-568) or 350 nm (DAPI) light sources. Images were captured on a Qicam CCD camera (QImaging) with a 100 ms exposure and false coloured ($\beta$III-tubulin, green; DAPI, blue). No $\beta$III-tubulin staining was observed when the primary antibody was omitted (data not shown).

## Image analysis

Images were analysed using ImageJ as previously described (Hartig, 2013). An automatic 'minimum error' threshold algorithm was applied to 8-bit images of $\beta$III-tubulin or DAPI staining to distinguish background from objects. Binary and raw images were manually compared, and the threshold manually adjusted to ensure all regions of interest were captured. The threshold was placed at the first minimum after the major peak of the image histogram. Binary images then underwent watershed segmentation to separate distinct objects in proximity. Identified particles, positive for either $\beta$III-tubulin or DAPI, were automatically counted using ImageJ and a ratio of $\beta$III-tubulin-positive cells (neurons) to DAPI-positive cells (neurons and satellite cells) calculated.

## *Ex vivo* electrophysiology recordings of colonic afferent activity

Conducted as previously described (Hockley et al., 2020), the distal colorectum and associated lumbar splanchnic nerve (LSN; rostral to inferior mesenteric ganglia) were isolated from mice euthanized as described above and cannulated in a rectangular recording chamber with Sylgard base (Dow Corning, UK). Colons were luminally perfused (200 $\mu$l/min) and serosally superfused (7 ml/min; 32−34°C) with carboxygenated Krebs buffer solution (in mM: 124 NaCl, 4.8 KCl, 1.3 NaH$_2$PO$_4$·H$_2$O, 2.5 CaCl$_2$·2H$_2$O, 1.2 MgSO$_4$·7H$_2$O, 11.1 D-(+)-glucose, and 25 NaHCO$_3$) supplemented with 10 $\mu$M atropine and 10 $\mu$M nifedipine to subject smooth muscle activity to neuromuscular blockade (Ness & Gebhart, 1988a).

Multi-unit activity from LSN bundles were recorded using borosilicate glass suction electrodes, and signals were amplified, band-pass filtered (gain 5 KHz; 100−1300 Hz; Neurolog, Digitimer Ltd, UK), and filtered digitally for 50 Hz noise (Humbug, Quest Scientific, Canada). Analogue signals were digitized at 20 kHz (Micro1401; Cambridge Electronic Design, UK). All signals were visualized using Spike2 software.

## Electrophysiology protocols

Following a minimum 30 min stabilization period, repeated ramp distensions (0–80 mmHg) of the colorectum were performed by occluding luminal perfusion outflow of the cannulated tissue (total distension taking approximately 220 s). Ramp distensions at pressures >30mmHg are noxious evoking pain behaviours in mice and humans (Hughes et al., 2009; Ness & Gebhart, 1988b).

In total, five ramp distensions were performed, separated by 15 min, after which 1 $\mu$M capsaicin (20 ml) was applied by bath superfusion (15 min after the last distension). TNFα (100 nM) or vehicle (buffer) was applied by luminal perfusion TNFα (15 min) was applied between the end of ramps 3 and 4. For experiments examining the effect of p38 MAPK inhibition, preparations were pre-treated by luminal perfusion with either SB203580 (10 $\mu$M) or vehicle (0.01% DMSO) started 15 min prior to and continued throughout TNFα luminal perfusion.

## Electrophysiological data analysis

In electrophysiological recordings, nerve discharge was determined by measuring the number of spikes passing a manually determined threshold twice the level of background noise (typically 60−80 $\mu$V) and binned to determine average firing frequency every 10 s. Changes in neuronal firing rates were calculated by subtracting baseline firing (averaged 3 min prior to distension or drug perfusion) from increases in nerve activity following ramp distension or capsaicin application. Peak firing to noxious mechanical distension and capsaicin application

was determined respectively as the highest neuronal activity during ramp distension 5 and during the 10 min post-capsaicin application. Changes to neuronal activity were recorded with each 5 mmHg increase in pressure and used to visualize ramp profiles. Capsaicin response profiles were plotted from binned data at 30 s increments. The area under the curve (AUC) was calculated for the duration of each ramp distension (0–80 mmHg) and for the 10 min following initial capsaicin application from response profiles using GraphPad Prism 9 software.

## Statistical analysis

All data sets were normality tested with a Shapiro–Wilk test and analysed using the appropriate statistical tests. The level of significance was set at $P \leq 0.05$. All data are displayed as means ± standard deviation. For $Ca^{2+}$ imaging analysis, $n$ represents the total number of dishes and $N$ represents the total number of mice from which they were cultured. On average, ~180 neurons were imaged per dish. In MACS cultures, $N$ represents the total number of independent pooled cultures.

## Results

### TNFα sensitizes TRPV1 signalling in DRG neurons

In keeping with studies showing TNFα potentiation of TRPV1-mediated currents and $Ca^{2+}$ flux in sensory neurons (Hsu et al., 2017; Khan et al., 2008), we examined the effect of overnight (24 h) incubation, or acute application of TNFα (3 nM), on capsaicin-evoked increases in intracellular $Ca^{2+}$ concentration ($[Ca^{2+}]_i$) within DRG sensory neurons. Overnight incubation with TNFα elicited a significantly greater peak increase in $[Ca^{2+}]_i$ to 10 s 1 μM capsaicin ($P = 0.028$, two-way ANOVA with Holm–Šídák's multiple comparisons; $n = 5$, $N = 5$; Fig. 1A and B) in a similar proportion of neurons compared with vehicle incubation ($P = 0.547$, two-way ANOVA with Holm–Šídák's multiple comparisons; $n = 5$, $N = 5$; Fig. 1C). No sensitization was observed at lower concentrations of capsaicin (0.01 μM: $P = 0.404$; 0.1 μM: $P = 0.404$, two-way ANOVA with Holm–Šídák's multiple comparisons; $n = 5$, $N = 5$) and the proportion of capsaicin-sensitive neurons was unchanged following incubation with TNFα (0.01 μM: $P = 0.901$; 0.1 μM: $P = 0.901$, two-way ANOVA with Holm–Šídák's multiple comparisons; $n = 5$, $N = 5$). Additionally, acute administration of TNFα (1 min), between repeat applications of capsaicin, prevented the marked desensitization of response to the second capsaicin application (Fig. 1D). This effect was only observed in a subset of capsaicin-responsive neurons (29.93 ± 11.76%) that were co-sensitive to TNFα and represented

82.15 ± 20.91% of total capsaicin responders (e.g. $cap_2/cap_1$ $P = 0.002$, $P = 0.004$, one-way ANOVA with Holm–Šídák's multiple comparisons test; $n = 5-6$, $N = 5-6$; Fig. 1E).

### TNFR1, TRPV1 and p38 MAPK signalling mediates the sensitization of DRG neurons by TNFα

Having confirmed that TNFα sensitizes responses to capsaicin in DRG neurons, we next investigated the rise in $[Ca^{2+}]_i$ elicited by 1 min TNFα alone. Application of TNFα elicited a concentration-dependent increase in the magnitude of response ($P = 0.050$, one-way ANOVA with Holm–Šídák's multiple comparisons test; $n = 5$, $N = 5$; Fig. 2A and B). Lower concentrations of TNFα did not elicit significantly different response magnitudes (0.03 nM *vs.* 0.1 nM: $P = 0.321$; 0.1 nM *vs.* 3 nM: $P = 0.321$, one-way ANOVA with Holm–Šídák's multiple comparisons test; $n = 5$, $N = 5$; Fig. 2A and B). TNFα activated a greater proportion of neurons at 3 nM compared with 0.03 nM ($P = 0.014$, the Kruskal–Wallis test with Dunn's multiple comparisons test; $n = 5$, $N = 5$; Fig. 2C). No difference in the proportion of TNFα-sensitive neurons was found at other concentrations (0.03 nM *vs.* 0.1 nM: $P > 0.999$; 0.1 nM *vs.* 3 nM: $P = 0.121$, the Kruskal–Wallis test with Dunn's multiple comparisons test; $n = 5$, $N = 5$; Fig. 2C). At the highest capsaicin concentration tested, 45.8 ± 9.37% of DRG neurons were activated by TNFα, of which 79.9 ± 10.7% were also co-sensitive for capsaicin, indicating a preferential activation of nociceptors by TNFα. In DRG neurons isolated from TNFR1$^{-/-}$ mice, the magnitudes of TNFα responses were significantly reduced ($P = 0.012$, one-way ANOVA with Holm–Šídák's multiple comparisons test, $n = 5-6$, $N = 5-6$; Fig. 2D and E) and the change in $[Ca^{2+}]_i$ in response to TNFα was no different to that produced by ECS (indicated by the dashed line in Fig. 2F; $P = 0.0001$, one-way ANOVA with Holm–Šídák's multiple comparisons test; $n = 5-6$, $N = 5-6$). Pre-treatment with R7050, an inhibitor of TNFR1 signalling (Cheng et al., 2021), attenuated the TNFα-mediated rise in $[Ca^{2+}]_i$ in DRG neurons isolated from wild-type (WT) mice ($P = 0.035$, one-way ANOVA with Holm–Šídák's multiple comparisons test; $n = 5$, $N = 5$; Fig. 2E) and reduced the proportion of TNFα-sensitive neurons ($P = 0.004$, one-way ANOVA with Holm–Šídák's multiple comparisons test; $n = 5$, $N = 5$; Fig. 2F). These observations confirm an essential role for TNFR1 in TNFα-mediated nociceptor stimulation.

Further experiments revealed that the TNFα-mediated increase in $[Ca^{2+}]_i$ was lost following removal of extracellular $Ca^{2+}$ ($P = 0.015$, the Kruskal–Wallis test with Dunn's multiple comparisons test; $n = 5-8$, $N = 5$; Fig. 3A), but unaffected by depletion of

intracellular $Ca^{2+}$ stores ($P > 0.999$, the Kruskal–Wallis test with Dunn's multiple comparisons test; $n = 5-8$, $N = 5$), demonstrating that the rise in $[Ca^{2+}]_i$ was driven by external $Ca^{2+}$ entry. This was further shown by a decrease in TNFα response magnitude following external $Ca^{2+}$ depletion ($P = 0.001$, the Kruskal–Wallis test with Dunn's multiple comparisons test; $n = 5-8$, $N = 5$; Fig. 3*B*), but no change in thapsigargin-treated neurons ($P = 0.310$, the Kruskal–Wallis test with Dunn's multiple comparisons test; $n = 5-8$, $N = 5$). Consistent with this observation, pre-treatment with

the TRPV1 antagonist A425619 (1 μM; Zhang et al., 2011) significantly attenuated the proportion of neurons responding to TNFα ($P = 0.037$, one-way ANOVA with Holm–Šídák's multiple comparisons test; $n = 5$, $N = 5$; Fig. 3*C*) at a concentration that abolished responses to capsaicin (e.g. $4.39 \pm 2.29\%$ in the presence of A425619 *vs.* $44.63 \pm 8.19\%$ in the absence of A425619; $P < 0.0001$, unpaired *t* test; $n = 4-5$, $N = 4-5$; data not shown). However, A425619 had no effect on the magnitude of responses in the population of neurons still activated by TNFα ($P = 0.431$, one-way ANOVA with Holm–Šídák's

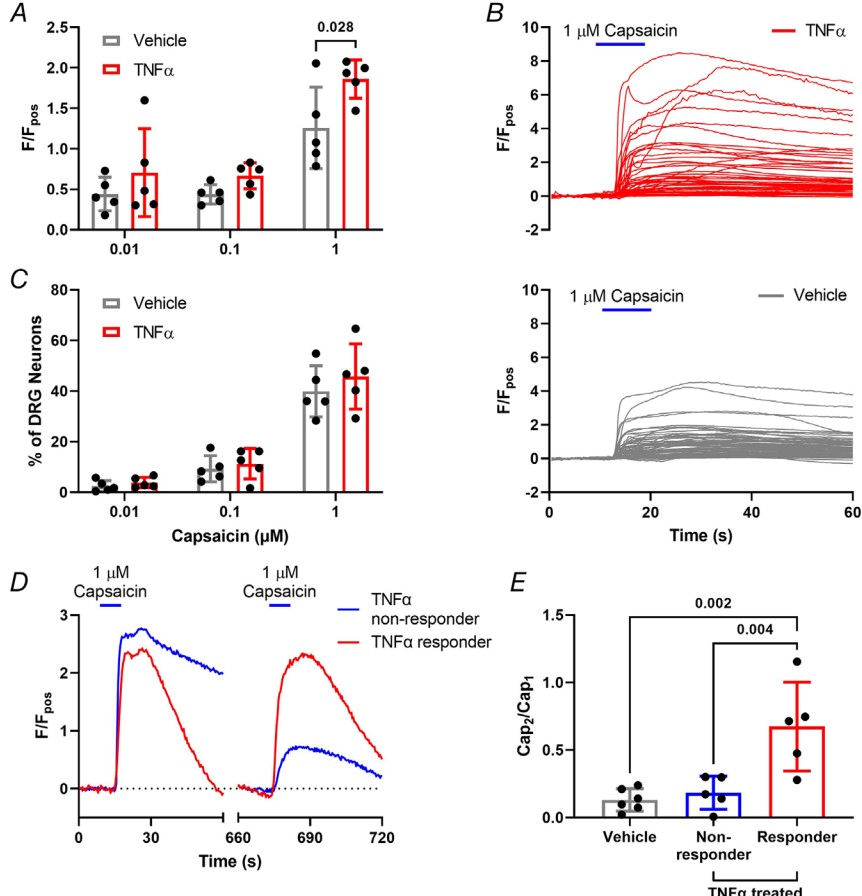

**Figure 1. TNFα sensitized capsaicin-evoked [Ca²⁺]ᵢ increase in dorsal root ganglion (DRG) neurons**
Capsaicin (1 μM) application resulted in (*A*) a significantly greater peak response averaged per dish in capsaicin-sensitive DRG neurons pre-incubated with TNFα compared with vehicle ($P = 0.028$, two-way ANOVA with Holm–Šídák's multiple comparisons test; $n = 5$ dishes from $N = 5$ mice per group). Representative response profiles illustrating (*B*) the greater effect of 1 μM capsaicin (10 s) on DRG neurons pre-incubated with TNFα (3 nM, top panel) compared with vehicle (PBS, bottom panel) for 24 h. *C*), the overall proportion of capsaicin-sensitive DRG neurons was comparable between culture dishes pre-incubated with vehicle or TNFα respectively at all capsaicin concentrations ($P = 0.775$, two-way ANOVA with Holm–Šídák's multiple comparisons test; $n = 5$, $N = 5$). In addition, the marked desensitization of responses to a repeat capsaicin application was greatly attenuated by TNFα, as illustrated (*D*) by the desensitizing response profile to repeat capsaicin application in a TNFα-insensitive (blue) DRG neuron in comparison with the lack of desensitization to capsaicin in a TNFα-sensitive (red) neuron; and confirmed in (*E*) by the significantly greater peak response ratios between the second (cap₂) and first (cap₁) capsaicin application in DRG neurons co-sensitive to TNFα (3 nM) compared with vehicle (ECS; $P = 0.002$) or TNFα insensitive neurons ($P = 0.004$, one-way ANOVA with Holm–Šídák's multiple comparisons test; $n = 5-6$ dishes from $N = 5-6$ independent cultures). [Colour figure can be viewed at wileyonlinelibrary.com]

multiple comparisons test; $n = 5$, $N = 5$; Fig. 3*D*). Furthermore, inhibition of TRPA1 with 1 $\mu$M AM0902 (Huang et al., 2019) also reduced the proportion of TNF$\alpha$ responders ($P = 0.001$, one-way ANOVA with Holm–Šídák's multiple comparisons test; $n = 5$, $N = 5$; Fig. 3*C*) and the magnitude of TNF$\alpha$ responses compared with controls ($P = 0.002$, one-way ANOVA with Holm–Šídák's multiple comparisons test; $n = 5$, $N = 5$; Fig. 3*D*, as well as attenuating TNF$\alpha$ Ca$^{2+}$ responses

more effectively than A425619 ($P = 0.019$, one-way ANOVA with Holm–Šídák's multiple comparisons test; $n = 5$, $N = 5$). Co-administration of A425619 and AM0902 to simultaneously inhibit TRPV1 and TRPA1 depleted the number of TNF$\alpha$-responsive neurons ($P < 0.0001$, one-way ANOVA with Holm–Šídák's multiple comparisons test; $n = 5$, $N = 5$; Fig. 3*C*) and decreased response magnitude compared with controls ($P = 0.003$, one-way ANOVA with Holm–Šídák's

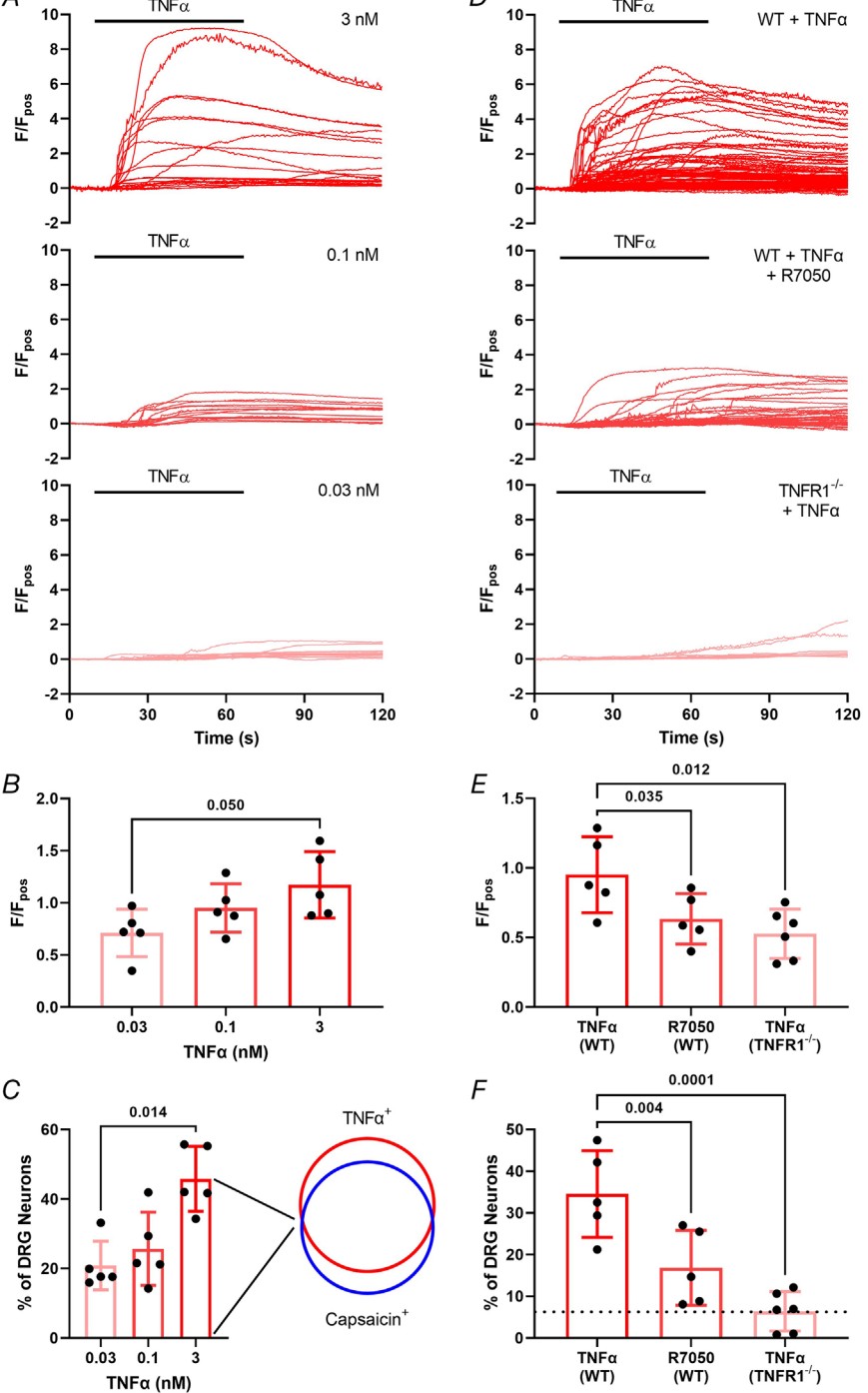

**Figure 2. Characterization of TNF$\alpha$-evoked [Ca$^{2+}$]$_i$ increase in dorsal root ganglion (DRG) neurons**

*A*, representative traces illustrating the concentration-dependent increase in the magnitude of the normalized fluorescent response to TNF$\alpha$ (0.03–3.0 nM) within individual neurons from respective culture dishes. *B*, the mean magnitude of TNF$\alpha$ responses per dish increased across TNF$\alpha$ concentrations ($P = 0.050$, one-way ANOVA with Holm–Šídák's multiple comparisons test; $n = 5$ dishes from $N = 5$ independent cultures). *C*, the concentration-dependent increase in the proportion (per dish) of TNF$\alpha$-responsive DRG neurons ($P = 0.014$, the Kruskal–Wallis test with Dunn's multiple comparisons test; $n = 5$ dishes from $N = 5$ independent cultures), of which the majority were also co-sensitive to capsaicin. *D*, the effects of TNF$\alpha$ (0.1 nM) were TNFR1-mediated as illustrated by the significant reduction in (*E*) response magnitude ($P = 0.035$; $P = 0.012$, one-way ANOVA with Holm–Šídák's multiple comparisons test, $n = 5-6$ dishes from $N = 5-6$ independent cultures) and (*F*) proportion of TNF$\alpha$-responsive neurons following TNFR1 inhibition with 10 $\mu$M R7050 ($P = 0.004$, one-way ANOVA with Holm–Šídák's multiple comparisons test; $n = 5-6$ dishes from $N = 5-6$ independent cultures) or genetic deletion of TNFR1 in Tnfrsf1a$^{-/-}$ mice in comparison with neurons from wild-type (WT) animals ($P = 0.0001$, one-way ANOVA with Holm–Šídák's multiple comparisons test; $n = 5-6$ dishes from $N = 5-6$ independent cultures). Dotted line represents the proportion of neurons activated in ECS controls (5.54 $\pm$ 5.01%, $n = 6$, $N = 6$). [Colour figure can be viewed at wileyonlinelibrary.com]

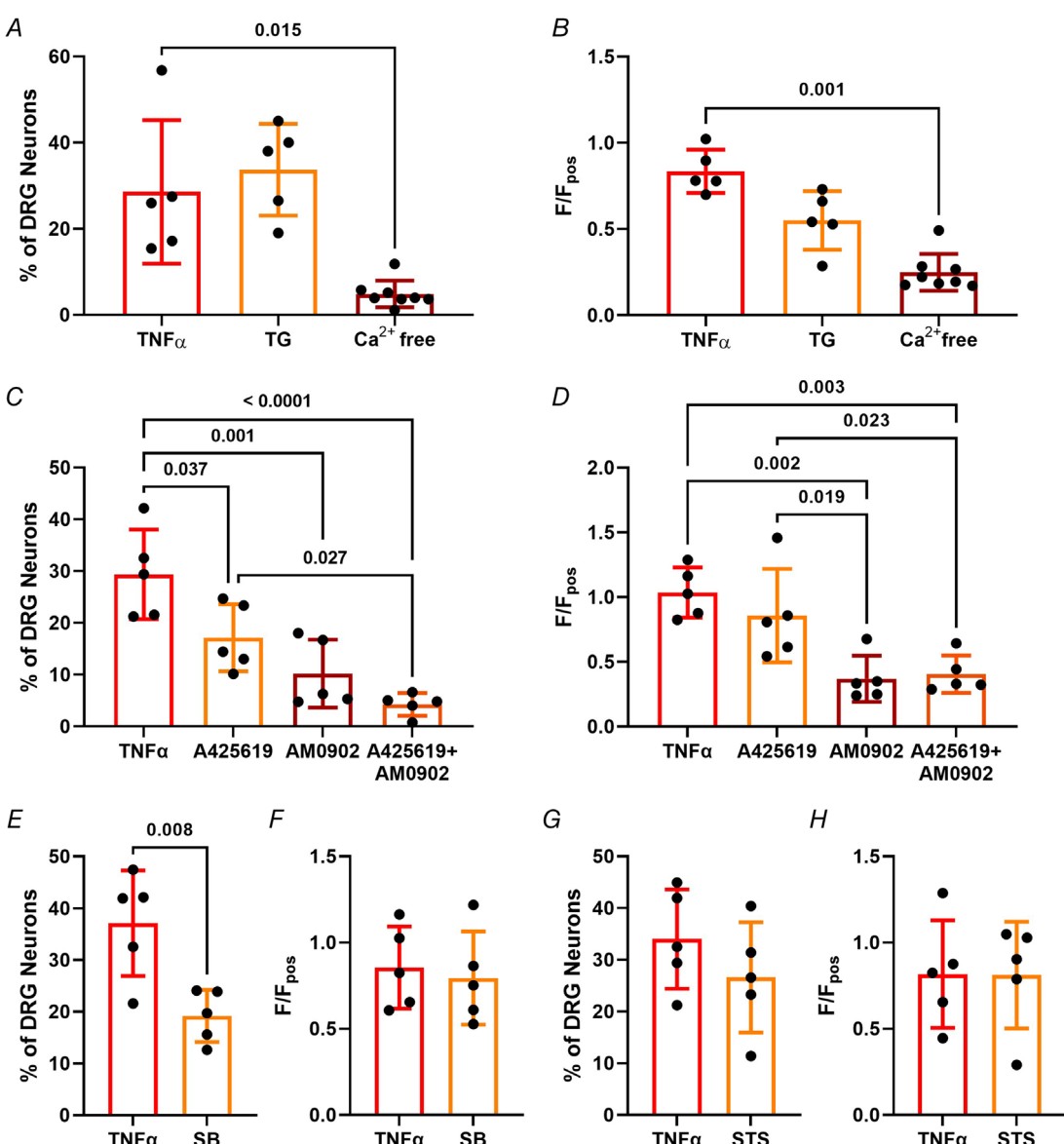

**Figure 3. TNFα-evoked [Ca²⁺]$_i$ increase is mediated by extracellular Ca²⁺, TRPV1 and p38 MAPK activation**

*A*, the proportion of dorsal root ganglion (DRG) neurons responding to 0.1 nM TNFα was unchanged following depletion of internal Ca²⁺ stores with 1 μM thapsigargin (TG; *P* > 0.999, the Kruskal–Wallis test with Dunn's multiple comparisons test; *n* = 5−8 dishes from *N* = 5 independent cultures) and abolished by removal of external Ca²⁺ (*P* = 0.015, the Kruskal–Wallis test with Dunn's multiple comparisons test; *n* = 5−8 dishes from *N* = 5 independent cultures). *B*, depletion of external Ca²⁺ also decreased TNFα response magnitude (*P* = 0.001, the Kruskal–Wallis test with Dunn's multiple comparisons test; *n* = 5−8 dishes from *N* = 5 independent cultures), but was unchanged by TG (*P* = 0.310, the Kruskal–Wallis test with Dunn's multiple comparisons test; *n* = 5−8 dishes from *N* = 5 independent cultures). This effect was partially mediated by TRPV1 and TRPA1 channels, as illustrated (*C*) by the additive reduction in TNFα-sensitive neurons following pre-incubation with TRPV1 inhibitor 1 μM A425619 and TRPA1 inhibitor 1 μM AM0902 (*P* < 0.0001, one-way ANOVA with Holm–Šídák's multiple comparisons test; *n* = 5 dishes from *N* = 5 independent cultures) and (*D*) the reduction in magnitude of Ca²⁺ response to TNFα (p 0.003, one-way ANOVA with Holm–Šídák's multiple comparisons test; *n* = 5 dishes from *N* = 5 independent cultures). Furthermore, (*E*) the proportion of neurons activated by TNFα was significantly attenuated following p38 MAPK inhibition with 10 μM SB203580 (SB; *P* = 0.008, unpaired *t* test; *n* = 5 dishes from *N* = 5 independent cultures), but (*F*) response magnitude was unaffected (*P* = 0.716, unpaired *t* test; *n* = 5 dishes from *N* = 5 independent cultures). In contrast, inhibition of protein kinase C (PKC) activity with 10 μM staurosporine (STS) had no effect on (*G*) the proportion of TNFα responders (*P* = 0.283, unpaired *t* test; *n* = 5 dishes from *N* = 5 independent cultures) or (*H*) the magnitude of TNFα responses (*P* = 0.977, unpaired *t* test; *n* = 5 dishes from *N* = 5 independent cultures). [Colour figure can be viewed at wileyonlinelibrary.com]

multiple comparisons test; $n = 5$, $N = 5$; Fig. 3D). A combination of A425619 and AM0902 resulted in a smaller proportion of TNFα-sensitive neurons compared with use of A425619 ($P = 0.027$, one-way ANOVA with Holm–Šídák's multiple comparisons test; $n = 5$, $N = 5$) and reduced TNFα responses ($P = 0.023$, one-way ANOVA with Holm–Šídák's multiple comparisons test; $n = 5$, $N = 5$). These results indicate a TRP-dependent component in TNFα-mediated neuronal activation.

In addition, pre-treatment with the p38 MAPK inhibitor SB203580 (10 μM; Wu et al., 2016), significantly reduced the proportion of TNFα-sensitive neurons ($P = 0.008$, unpaired $t$ test; $n = 5$, $N = 5$; Fig. 3E) in agreement with previous findings demonstrating a role for p38 MAPK in TNFα-mediated neuronal sensitization (Gudes et al., 2015). SB203580 had no significant effect on TNFα response magnitude ($P = 0.716$, unpaired $t$ test; $n = 5$, $N = 5$; Fig. 3F). In contrast, the protein kinase C (PKC) inhibitor staurosporine (10 μM; Rusin & Moises, 1998) failed to reduce the proportion of TNFα-sensitive neurons ($P = 0.283$, unpaired $t$ test; $n = 5$, $N = 5$; Fig. 3G) or response magnitude ($P = 0.977$, unpaired $t$ test; $n = 5$, $N = 5$; Fig. 3H).

Furthermore, in ultra-pure DRG neuron cultures, in which non-neuronal cells were removed by MACS ($P < 0.0001$, unpaired $t$ test; $n = 3$, $N = 3$; Fig. 4A and B), the response to TNFα (Fig. 4C) was still observed in a comparable proportion of neurons ($P = 0.491$, unpaired $t$ test; $n = 4-5$, $N = 4-5$; Fig. 4D), and the magnitude of TNFα responses was unchanged ($P = 0.426$, unpaired $t$ test; $n = 4-5$, $N = 4-5$; Fig. 4E), thereby confirming that TNFα can directly stimulate sensory neurons, consistent with reported TNFR1 expression in DRG neurons. In ultra-pure neuronal cultures, MACS reduced the number of large diameter neurons, leaving a preparation enriched in nociceptors, as seen in previous studies (Thakur et al., 2014) (Fig. 4A).

Having established the important contribution of p38 MAPK signalling and TNFR1 expression to TNFα-mediated $Ca^{2+}$ flux, we next confirmed the involvement of this pathway in the sensitization of TRPV1 signalling by TNFα. No sensitization of the magnitude of the $[Ca^{2+}]_i$ response to capsaicin following 24 h incubation with TNFα was observed in tissue from TNFR1$^{-/-}$ mice ($P = 0.787$, unpaired $t$ test; $n = 6$, $N = 6$; Fig. 5A and B), and the proportion of capsaicin-sensitive DRG neurons was comparable to controls ($P = 0.891$, unpaired $t$ test; $n = 6$, $N = 6$; Fig. 5C). Following co-incubation of TNFα with SB203580, TNFα-mediated sensitization of capsaicin responses was attenuated compared with vehicle and SB203580 controls ($P = 0.910$; $P = 0.944$, one-way ANOVA with Holm–Šídák's multiple comparisons test; $n = 5$, $N = 5$; Fig. 5D and E), and the proportion of responders unchanged ($P = 0.380$;

$P = 0.902$, one-way ANOVA with Holm–Šídák's multiple comparisons test; $n = 5$, $N = 5$; Fig. 5F).

## TNFα sensitizes colonic afferent responses to noxious ramp distension and capsaicin via TNFR1 and p38 MAPK

Finally, to confirm the translation of our findings from DRG neurons to the activation of colonic afferents, we studied the contribution of TNFR1 and p38 MAPK signalling to TNFα-mediated sensitization of colonic afferent responses to noxious ramp distension and capsaicin (Fig. 6A). Consistent with the sensitizing effects of TNFα observed previously in DRG neurons, treatment with TNFα-sensitized colonic afferent responses to repeated ramp distensions compared with vehicle ($P = 0.0004$, two-way ANOVA with Holm–Šídák's multiple comparisons test; $N = 8$; Fig. 6B). Comparisons within treatment groups demonstrated a significant increase in afferent response between ramp distensions 3 and 5 in TNFα-treated tissues ($P = 0.011$, two-way ANOVA with Holm–Šídák's multiple comparisons test; $N = 8$), whereas no significant change occurred between responses to ramp distensions 3 and 5 in vehicle-treated tissues ($P = 0.425$, two-way ANOVA with Holm–Šídák's multiple comparisons test; $N = 8$). Significantly greater afferent responses to ramp distension were observed across noxious distending pressures ($P = 0.0002$, multiple unpaired $t$ tests, $N = 8$; Fig. 6C) and peak afferent firing was significantly increased ($P = 0.0004$, unpaired $t$ test, $N = 8$; Fig. 6D). An enhanced afferent response to capsaicin was also observed following application of TNFα compared with vehicle ($P = 0.003$, Mann–Whitney's test; $N = 8$; Fig. 7A and B). TNFα significantly increased nerve firing in the 10 min post-capsaicin application ($P = 0.005$, two-way ANOVA with Holm–Šídák's multiple comparisons test; $N = 8$; Fig. 7C) and peak activity was elevated 75% compared with vehicle ($P = 0.007$, unpaired $t$ test, $N = 8$; Fig. 7D). Compliance of distensions was comparable between treatment groups (e.g. $0.741 \pm 0.137$ ml *vs.* $0.731 \pm 0.074$ ml for the fifth ramp distension in TNFα *vs.* vehicle-treated tissues, $P = 0.842$, unpaired $t$ test; $N = 8$; data not shown).

In keeping with data from DRG neurons, where we observed a reliance on TNFR1 for the sensitizing effects of TNFα, no sensitization of responses to noxious ramp distension was observed following the application of TNFα compared with vehicle in tissue from TNFR1$^{-/-}$ mice ($P = 0.521$, unpaired $t$ test; $N = 7-8$; Fig. 8A). In addition, no difference in afferent activity was seen across noxious distending pressures ($P = 0.280$, multiple unpaired $t$ tests; $N = 7-8$; Fig. 8B) and peak afferent firing was unchanged between vehicle- and TNFα-treated

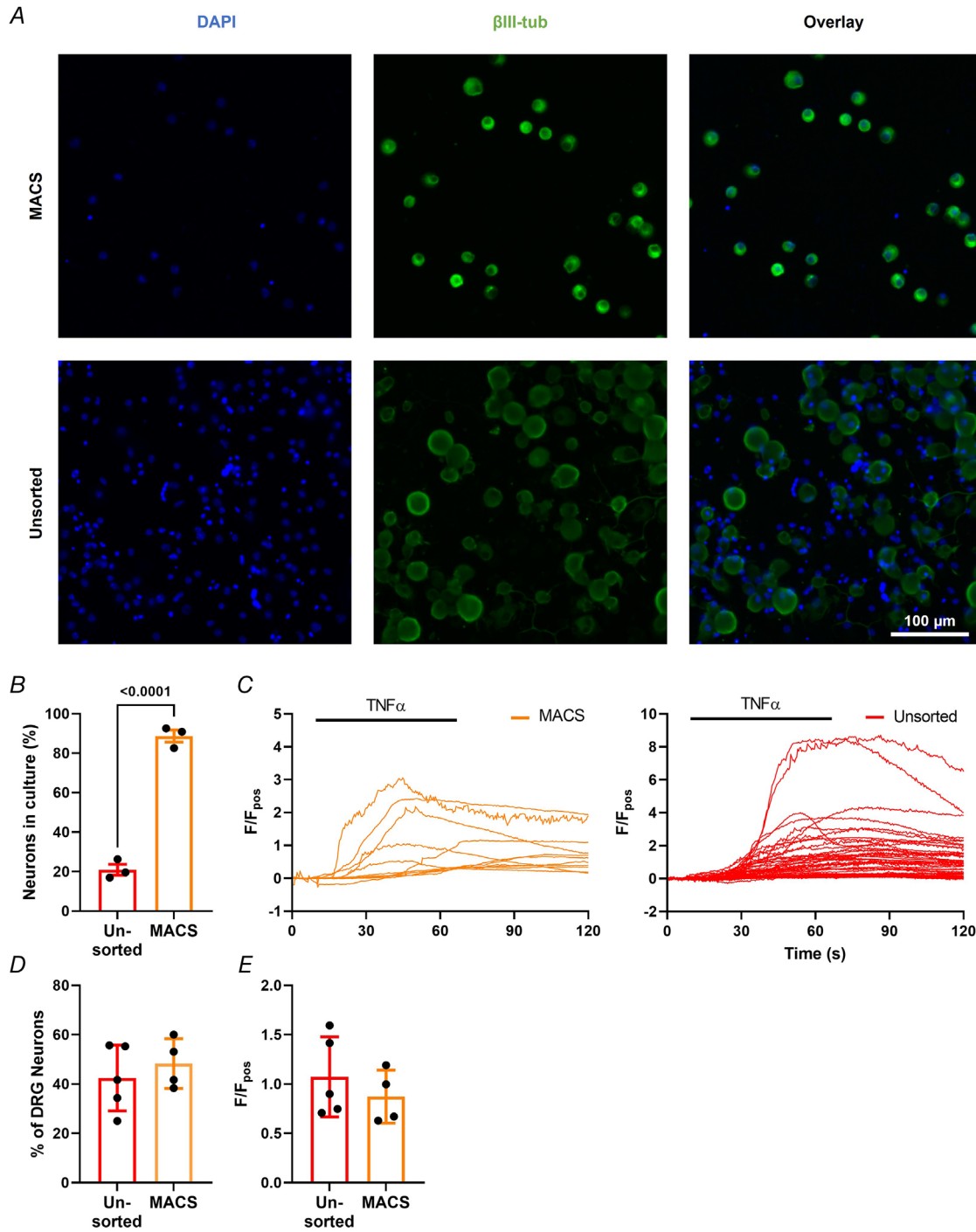

**Figure 4. TNFα acts directly on dorsal root ganglion (DRG) neurons**
Magnetic-activated cell sorting (MACS) was used to generate ultra-pure neuronal cultures as illustrated by (*A*) the immunofluorescent staining of neuronal marker βIII-tubulin (green) and nuclear DAPI stain (blue) in respective cultures and (*B*) the greatly increased proportion of cells stained with βIII-tubulin following MACS sorting ($P < 0.0001$, unpaired *t* test; $n = 3$ dishes from $N = 3$ pooled cultures). TNFα mediated robust increases in $[Ca^{2+}]_i$ in ultra-pure and unsorted cultures illustrated (*C*) by individual traces from respective cultures and confirmed by (*D*) the comparable percentage of TNFα responders in unsorted and MACS-sorted DRG cultures ($P = 0.491$, unpaired *t* test; $n = 4–5$ dishes from $N = 4–5$ pooled cultures) and (*E*) equivocal peak responses per dish ($P = 0.426$, unpaired *t* test; $n = 4–5$ dishes from $N = 4–5$ pooled cultures).

TNFR1$^{-/-}$ tissue ($P = 0.666$, unpaired $t$ test; $N = 7-8$; Fig. 8*C*). No TNF$\alpha$-mediated sensitization was observed in response to capsaicin ($P = 0.767$, unpaired $t$ test; $N = 7-8$; Fig. 8*D*), and afferent firing was comparable 10 min after capsaicin application ($P = 0.961$, two-way ANOVA with Holm–Šídák's multiple comparisons test; $N = 7-8$; Fig. 8*E*). Furthermore, TNF$\alpha$ had no effect on peak afferent activity in the absence of TNFR1 ($P = 0.967$, unpaired $t$ test; $N = 7-8$; Fig. 8*F*). Interestingly, responses to ramp distension (AUC: WT $769 \pm 312$ *vs.* TNFR1$^{-/-}$ $1665 \pm 412$; $P = 0.0004$, unpaired $t$ test; $N = 7-8$)

and capsaicin (AUC: WT $2114 \pm 1335$ *vs.* TNFR1$^{-/-}$ $5615 \pm 2535$; $P = 0.005$, unpaired $t$ test; $N = 7-8$) were significantly higher in vehicle-treated tissues from TNFR1$^{-/-}$ mice than WT mice.

In agreement with our identification of a role for p38 MAPK signalling in the sensitizing effects of TNF$\alpha$ in DRG neurons, we observed no sensitization of colonic afferents to noxious ramp distension following the application of TNF$\alpha$ in the presence of SB203580 to inhibit p38 MAPK ($P = 0.042$, the Kruskal–Wallis test with Dunn's multiple comparisons test; $N = 6-8$;

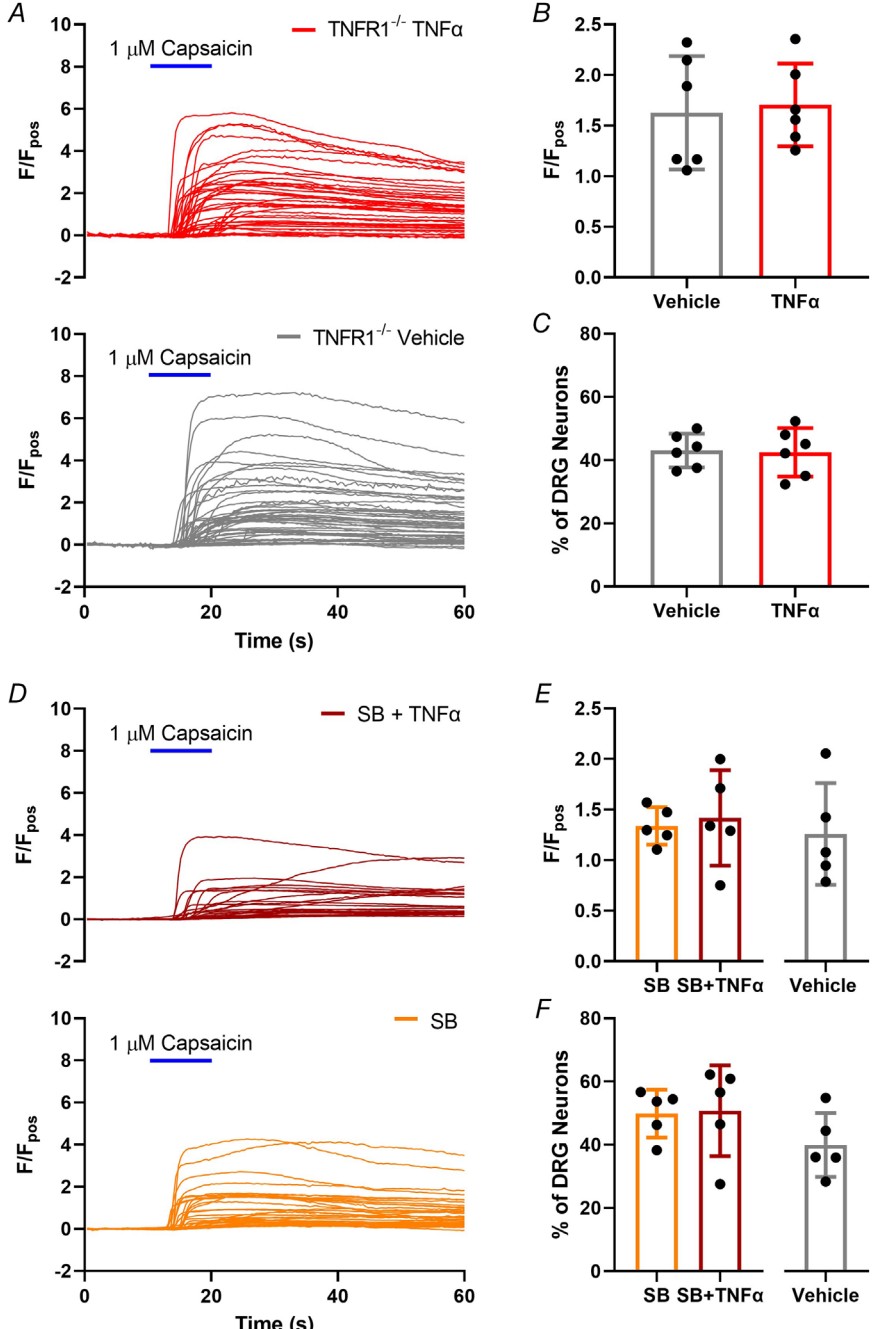

**Figure 5. TNF$\alpha$ sensitization of capsaicin-evoked [Ca$^{2+}$]$_i$ increase in dorsal root ganglion (DRG) neurons is p38 MAPK- and TNFR1-mediated**

*A*, example traces of individual response profiles to 1 $\mu$M capsaicin (10 s) in DRG neurons cultured from TNFR1$^{-/-}$ mice following 24 h treatment of respective culture dishes with vehicle (PBS) or TNF$\alpha$ (3 nM). *B*, incubation with TNF$\alpha$ no longer increased peak averaged per dish Ca$^{2+}$ responses to capsaicin in DRG neurons from TNFR1$^{-/-}$ mice ($P = 0.787$, unpaired $t$ test; $n = 6$ dishes from $N = 6$ independent cultures). *C*, the proportion of capsaicin-sensitive neurons remained comparable between culture dishes pre-incubated with vehicle or TNF$\alpha$ in DRG neurons from TNFR1$^{-/-}$ mice ($P = 0.891$, unpaired $t$ test; $n = 6$ dishes from $N = 6$ independent cultures). Similarly, overnight incubation with TNF$\alpha$ no longer increased the magnitude of capsaicin responses in DRG neurons co-incubated with the p38 MAPK inhibitor SB203580 as illustrated by: *D*, individual neuronal responses to 1 $\mu$M capsaicin (10 s) in SB203580-treated DRG neurons following 24 h incubation with vehicle or TNF$\alpha$ (3 nM); *E*, the comparable magnitude of the peak averaged per dish response to capsaicin between respective treatments ($P = 0.944$, one-way ANOVA with Holm–Šídák's multiple comparisons test; $n = 5$ dishes from $N = 5$ independent cultures). *F*, the proportion of capsaicin-sensitive DRG neurons was also comparable between treatment groups following p38 MAPK inhibition ($P = 0.380$, one-way ANOVA with Holm–Šídák's multiple comparisons test; $n = 5$ dishes from $N = 5$ independent cultures). [Colour figure can be viewed at wileyonlinelibrary.com]

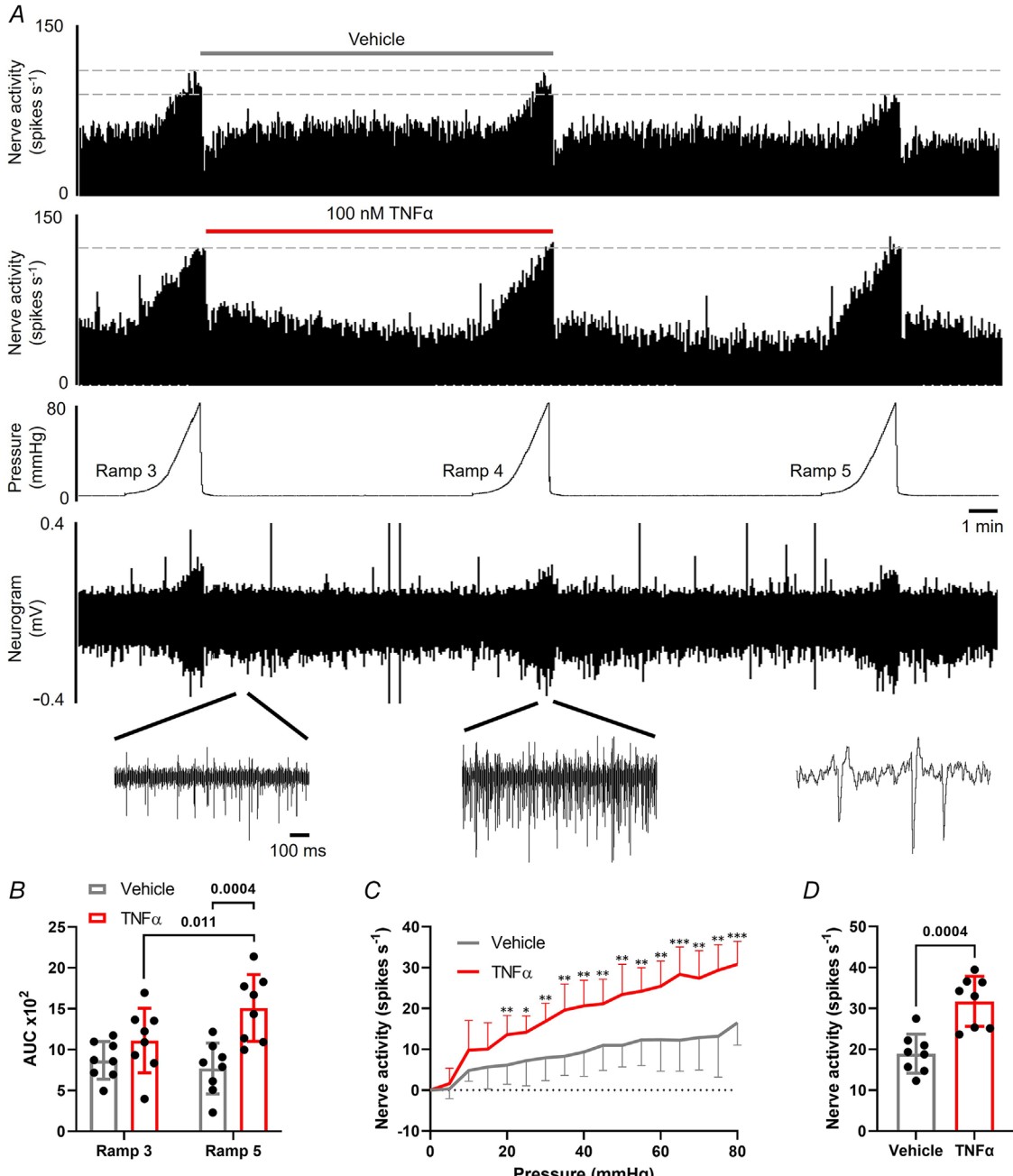

**Figure 6. TNFα evokes colonic afferent mechanical hypersensitivity**

*A*, example rate histograms and neurogram of lumbar splanchnic nerve (LSN) activity with accompanying pressure trace showing sequential (x3) ramp distensions (0–80 mmHg) from vehicle- and TNFα (100 nM)-treated preparations, highlighting the sensitization of afferent responses to distension following TNFα treatment. This effect was confirmed by: *B*, the significantly greater afferent response to ramp distension (measured by area under the curve, AUC, during ramp distension) following TNFα pre-treatment compared with vehicle (ramp 5) (*P* = 0.0004, two-way ANOVA with Holm–Šídák's multiple comparisons test; *N* = 8) and within treatment groups (*P* = 0.011; two-way ANOVA with Holm–Šídák's multiple comparisons test; *N* = 8); *C*, the significantly greater increase in afferent response throughout the noxious distending pressure range (≥20 mmHg, ramp 5) following TNFα treatment compared with vehicle (**P* < 0.05, ***P* < 0.01, ****P* < 0.001, multiple *t* tests; *N* = 8 animals); *D*, the significantly greater peak increase in afferent discharge to ramp distension following application of TNFα compared with vehicle (*P* = 0.0004, unpaired *t* test; *N* = 8). [Colour figure can be viewed at wileyonlinelibrary.com]

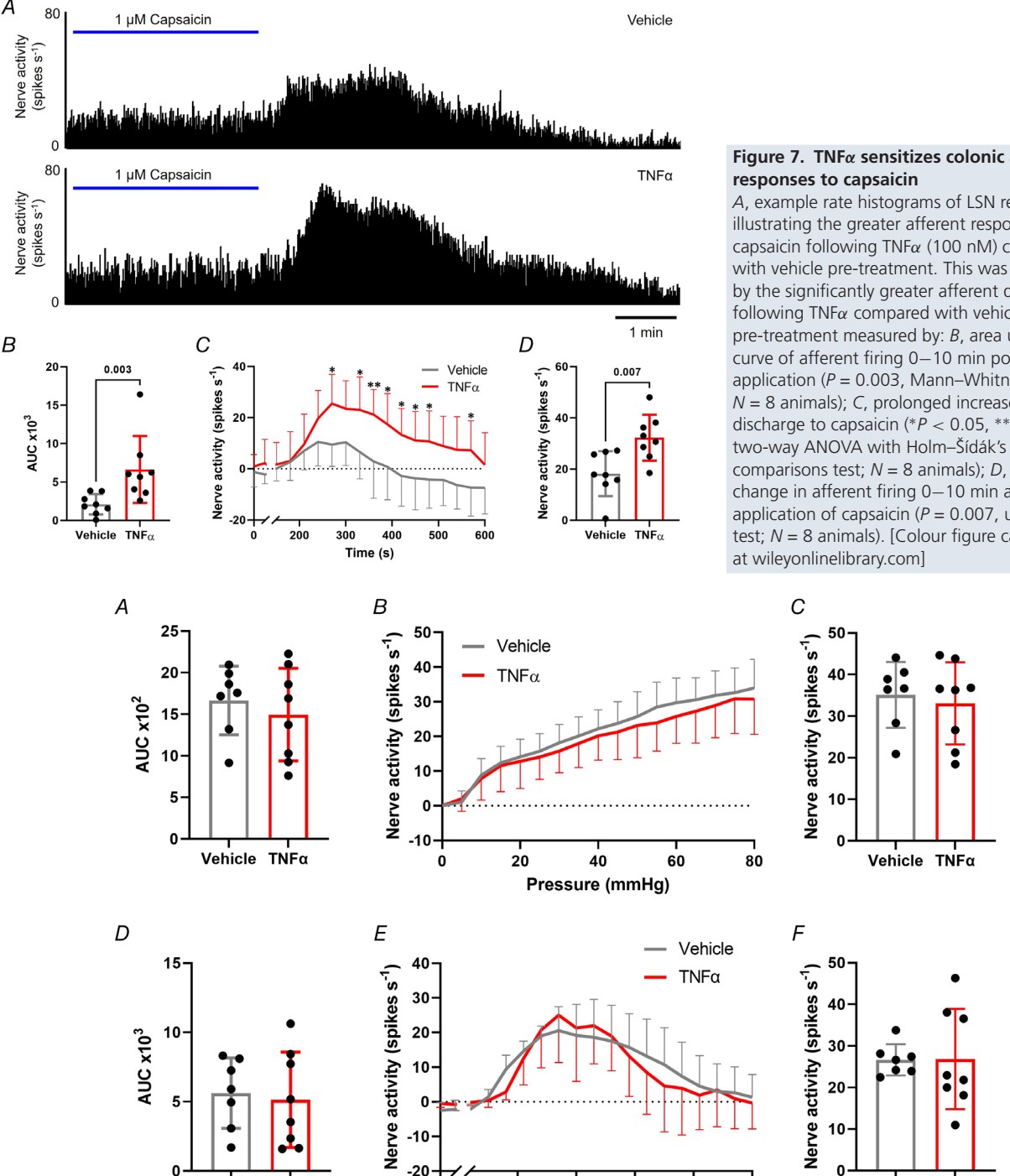

**Figure 7. TNFα sensitizes colonic afferent responses to capsaicin**

*A*, example rate histograms of LSN recordings illustrating the greater afferent response to 1 $\mu$M capsaicin following TNFα (100 nM) compared with vehicle pre-treatment. This was confirmed by the significantly greater afferent discharge following TNFα compared with vehicle pre-treatment measured by: *B*, area under the curve of afferent firing 0−10 min post capsaicin application ($P = 0.003$, Mann–Whitney's test; $N = 8$ animals); *C*, prolonged increase in afferent discharge to capsaicin (*$P < 0.05$, **$P < 0.01$, two-way ANOVA with Holm–Šídák's multiple comparisons test; $N = 8$ animals); *D*, greater peak change in afferent firing 0−10 min after application of capsaicin ($P = 0.007$, unpaired *t* test; $N = 8$ animals). [Colour figure can be viewed at wileyonlinelibrary.com]

**Figure 8. TNFR1 mediates TNFα-induced colonic afferent sensitization**

The sensitization of colonic afferents by TNFα pre-treatment was dependent on TNFR1 expression as confirmed by the presence of comparable colonic afferent responses to ramp distension and capsaicin following vehicle or TNFα pre-treatment in tissue from TNFR1$^{−/−}$ mice shown by: *A*, comparing the area under the curve (AUC) of nerve activity following ramp distension ($P = 0.521$, unpaired *t* test; $N = 7−8$); *B*, afferent response profiles to ramp distension ($P = 0.280$, multiple *t* tests; $N = 7−8$ animals); *C*, peak firing frequency to ramp distension ($P = 0.666$, unpaired *t* test; $N = 7−8$); *D*, AUC of firing frequency following capsaicin application ($P = 0.767$, unpaired *t* test; $N = 7−8$); *E*, afferent response profiles to capsaicin ($P = 0.961$, two-way ANOVA with Holm–Šídák's multiple comparisons test; $N = 7−8$ animals); *F*, the peak increase in afferent discharge to capsaicin ($P = 0.967$, unpaired *t* tests; $N = 7−8$). [Colour figure can be viewed at wileyonlinelibrary.com]

Fig. 9*A*) or in SB203580 controls (*P* = 0.011, the Kruskal–Wallis test with Dunn's multiple comparisons test; *N* = 6−8). SB203580 attenuated TNFα sensitization across all noxious distending pressures (*P* = 0.004; multiple unpaired *t* tests; *N* = 7−8; Fig. 9*B*) and p38 MAPK inhibition significantly decreased peak afferent firing following TNFα incubation (*P* = 0.026, the Kruskal–Wallis test with Dunn's multiple comparisons test; *N* = 6−8; Fig. 9*C*). As expected, no sensitization was observed in SB203580 controls (*P* = 0.020, the Kruskal–Wallis test with Dunn's multiple comparisons test; *N* = 6−8; Fig. 9*C*). Furthermore, SB203580 significantly attenuated afferent responses to capsaicin in TNFα-treated tissues (*P* = 0.034, the Kruskal–Wallis test with Dunn's multiple comparisons test; *N* = 7−8; Fig. 9*D*), with significantly reduced nerve activity profiles (*P* = 0.0002, two-way ANOVA with Holm–Šídák's multiple comparisons test; *N* = 7−8; Fig. 9*E*). Responses to capsaicin in SB203580 controls were significantly

lower than TNFα-treated tissues (*P* = 0.017, the Kruskal–Wallis test with Dunn's multiple comparisons test; *N* = 7−8; Fig. 9*D*). Additionally, SB203580 reversed the TNFα-mediated increase in peak afferent firing (*P* = 0.022, one-way ANOVA with Holm–Šídák's multiple comparisons test; *N* = 7−8; Fig. 9*F*) and as expected peak firing was unchanged in SB203580 controls (*P* = 0.022, one-way ANOVA with Holm–Šídák's multiple comparisons test; *N* = 7−8; Fig. 9*F*).

## Discussion

TNFα has been linked to the production of abdominal pain in gastrointestinal disease due to its enhanced expression in disease states such as IBS (Hughes et al., 2013) and IBD (Kamada et al., 2008), combined with data showing TNFα-mediated visceral nociceptor sensitization and TNFR1 expression in colonic nociceptors (Hockley et al., 2019). The signalling pathways mediating these

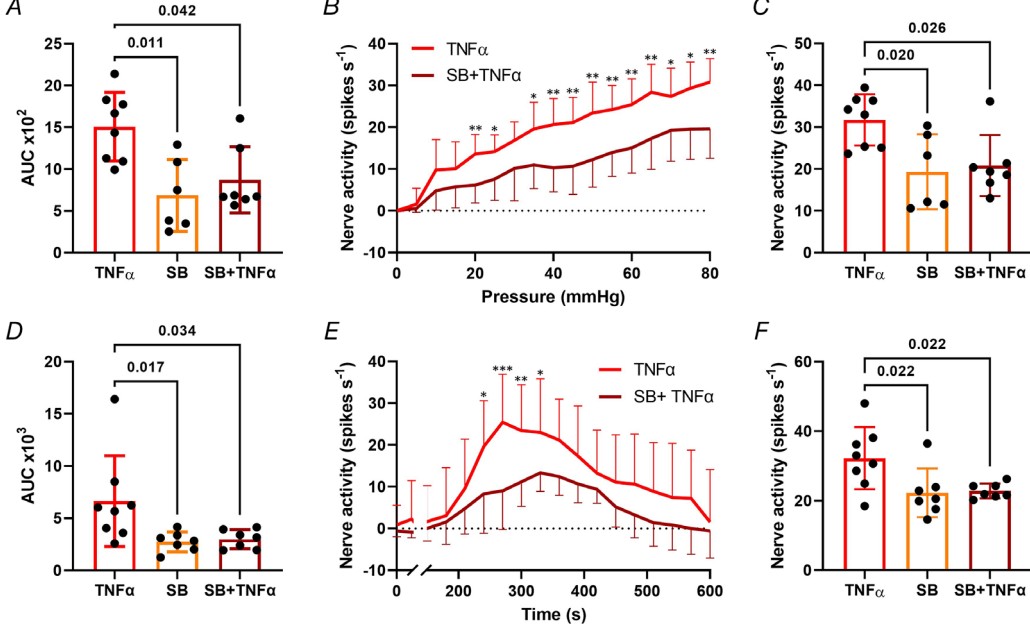

**Figure 9. TNFα-induced colonic afferent sensitization is mediated by p38 MAPK**
The ability of TNFα to sensitize colonic afferent responses to ramp distension and capsaicin was abolished by co-administration with the p38 MAPK inhibitor SB203589 as demonstrated by: *A*, the significant reduction in area under the curve (AUC) of colonic afferent responses to ramp distensions following pre-treatment with TNFα in the presence of SB203589 (*P* = 0.042, the Kruskal–Wallis test with Dunn's multiple comparisons test; *N* = 6−8 animals); *B*, the significant reduction in afferent responses to ramp distension across noxious distending pressures (≥20 mmHg) following pre-treatment with TNFα in the presence of SB203580 (\**P* < 0.05, \*\**P* < 0.01, multiple *t* tests; *N* = 6−8 animals); *C*, the significant reduction in the peak afferent response to ramp distension following application of TNFα in the presence of SB203580 (*P* = 0.026, the Kruskal–Wallis test with Dunn's multiple comparisons test; *N* = 6−8 animals); *D*, significantly reduced afferent responses to capsaicin measured by AUC following pre-treatment with TNFα in the presence of SB203580 (*P* = 0.034, the Kruskal–Wallis test with Dunn's multiple comparisons test; *N* = 7−8 animals); *E*, the significant reduction in afferent responses to capsaicin at multiple time points following pre-treatment with TNFα in the presence of SB203580 (\**P* < 0.05, \*\**P* < 0.01, \*\*\**P* < 0.001, two-way ANOVA with Holm–Šídák's multiple comparisons test; *N* = 7−8 animals); *F*, the significant reduction in peak nerve discharge to capsaicin following pre-treatment with TNFα in the presence of SB203580 (*P* = 0.022, one-way ANOVA Holm–Šídák's multiple comparisons test; *N* = 7−8 animals). [Colour figure can be viewed at wileyonlinelibrary.com]

effects have not been fully explored, although data from studies of somatic hypersensitivity point to a role for p38 MAPK signalling downstream of TNFR1 receptor activation (Jin & Gereau IV, 2006) and a sensitizing effect on TRPV1, a detector of noxious stimuli (Khan et al., 2008). As such, the goal of this study was to investigate the contribution of TNFR1 and p38 MAPK signalling to the pro-nociceptive effects of TNFα including TRPV1 receptor activation in sensory neurons and colonic afferents.

Data from our studies confirmed that TNFα sensitizes TRPV1 receptor signalling, by demonstrating: (i) an enhanced capsaicin-evoked increase in $[Ca^{2+}]_i$ within sensory DRG neurons following overnight incubation with TNFα; (ii) a TRPV1 receptor-mediated increase in $[Ca^{2+}]_i$ to acute application of TNFα in sensory DRG neurons that subsequently showed significantly less desensitization to repeated application of capsaicin; and (iii) a marked increase in the colonic afferent response to capsaicin following TNFα pre-treatment. In keeping with our hypothesis, these effects were no longer observed in tissue from TNFR1$^{-/-}$ mice and were greatly attenuated by pre-treatment with the p38 MAPK inhibitor SB203580; conditions in which TNFα also no longer prevented the desensitization of colonic afferent responses to repeated noxious distension of the bowel. Collectively these findings provide further functional evidence of a contribution by TNFα to the production of visceral pain in gastrointestinal disease and highlight the role of TNFR1-mediated p38 MAPK signalling to the pro-nociceptive activity of TNFα.

The validity of our data is supported by comparable observations showing TNFα enhances TRPV1 receptor signalling in nodose (Hsu et al., 2017; Lin et al., 2017), trigeminal (Khan et al., 2008) and DRG neurons (Hensellek et al., 2007). These effects have been attributed to TNFR1 activation and p38 MAPK signalling, consistent with our data demonstrating the essential contribution of TNFR1 and p38 MAPK to TNFα sensitization of capsaicin responses in colonic afferents for the first time, and DRG neurons for the first time in the same study. The mechanism by which p38 MAPK modulates TRPV1 receptor activity was not a focus of our experiments; however, we observed changes following brief application of TNFα indicative of an acute increase in channel activity that would be expected following phosphorylation of TRPV1 by p38 MAPK. This observation was in keeping with previous studies showing enhanced TRPV1-mediated inward currents shortly after application of TNFα; effects reported to be mediated by p38 MAPK and protein kinase C (PKC) (Constantin et al., 2008). However, in contrast to p38 MAPK inhibition, we found no change in the TRPV1-mediated increase in $[Ca^{2+}]_i$ to TNFα following PKC inhibition with staurosporine. These findings suggest

that phosphorylation sites on intracellular domains of TRPV1, which are not specific to PKC, may be targeted and phosphorylated by p38 MAPK. In addition, p38 MAPK signalling has also been shown to increase TRPV1 channel expression following incubation with TNFα for periods of 1 h or greater (Constantin et al., 2008), and this process may also have contributed to the increased capsaicin response we observed following overnight incubation with TNFα.

Consistent with the reported co-expression of TNFR1 with TRPV1 in sensory DRG neurons (Zeisel et al., 2018), we found marked co-sensitivity of TNFα-stimulated DRG neurons to capsaicin. This response was dependent on TNFR1 expression and was preserved in ultra-pure, neuron-only DRG cultures demonstrating that TNFα can directly modulate sensory neuron activity and TRPV1 signalling via TNFR1 in the same neuron. Furthermore, these data indicate that TNFα-responsive neurons are predominantly nociceptors based on the ability of capsaicin to evoke significant pain in humans, including visceral pain following colorectal administration (van Wanrooij et al., 2014).

This observation was corroborated by our finding that TNFα sensitizes colonic afferents at noxious distending pressures. These experiments provide evidence that TNFα causes abdominal pain through TNFR1-mediated colonic nociceptor sensitization due to p38 MAPK-enhanced TRPV1 signalling.

However, this is not the only mechanism by which TNFα stimulates sensory neurons. For example, the increase in $[Ca^{2+}]_i$ to TNFα, which is solely dependent on extracellular $Ca^{2+}$ entry, was only partially blocked by the TRPV1 antagonist (A425619) at concentrations that completely abolished the $Ca^{2+}$ response to capsaicin. This indicates that other $Ca^{2+}$ permeable ion channels are stimulated by TNFα and the additive inhibitory effect of the TRPA1 antagonist (AM0902) on TNFα-mediated neuronal activation is consistent with existing data implicating TRPA1 in TNFα-mediated afferent sensitization (Hughes et al., 2013). In addition, we also demonstrated that TNFα-mediated colonic afferent mechanosensitization was dependent on TNFR1 expression and p38 MAPK activity in keeping with seminal data showing this signalling pathway mediates somatic mechanical hypersensitivity to TNFα (Jin & Gereau IV, 2006). These effects have been attributed to enhanced TTX-resistant Na$_V$1.8 currents, an effect also seen in response to TNFR1-dependent TNFα signalling in colon projecting DRG neurons. With the inclusion of our study, current data indicate that TNFα has the capacity to sensitize colonic nociceptors through p38 MAPK-enhanced TRPV1, TRPA1 and Na$_V$1.8 channel activity downstream of TNFR1.

These findings highlight the utility of TNFα and its downstream signalling pathways as important drug

targets for pain relief in gastrointestinal diseases, such as IBS and IBD, in which enhanced TNF*α* expression has been reported.

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

## Additional information

### Data availability statement

All data supporting the results presented in the manuscript are included in the manuscript figures and raw data sets are available at DOI: https://doi.org/10.17863/CAM.81112.

### Competing interests

K.H.B. is supported by an AstraZeneca PhD Studentship. F.W. and I.P.C. are employed by AstraZeneca. E.St.J.S. and D.C.B. receive research funding from AstraZeneca.

### Author contributions

K.H.B. designed the research studies, conducted the experiments, acquired and analysed the data and wrote the manuscript. J.P.H. acquired and analysed the data and wrote the manuscript. T.S.T. acquired the data. L.A.P. analysed the data. I.P.C. designed the research studies. D.C.B., E.St.J.S. and F.W. designed the research studies and wrote the manuscript. All authors approved the final version of the manuscript submitted for publication and agree to be accountable for all aspects of the work in ensuring that questions related to the accuracy or integrity of any part of the work are appropriately investigated and resolved. All persons designated as authors qualify for authorship, and all those who qualify for authorship are listed.

### Funding

This work was supported by an AstraZeneca PhD Studentship (K.H.B.: RG98186) and Biotechnology and Biological Sciences Research Council Doctoral Training Program awarded to the University of Cambridge (J.P.H. & L.A.P.: BB/M011194/1).

### Keywords

capsaicin, inflammatory bowel disease, irritable bowel syndrome, nociception, p38 MAPK, tumour necrosis factor-alpha, TRPV1, visceral pain

## Supporting information

Additional supporting information can be found online in the Supporting Information section at the end of the HTML view of the article. Supporting information files available:

**Statistical Summary Document**
**Peer Review History**

