## [Peer Review History · The Journal of Physiology]

Sensitisation of colonic nociceptors by TNF α is dependent on TNFR1 expression and p38 MAPK activity

Katie H Barker, James P Higham, Luke A Pattison, Toni Stacey Taylor, Iain P Chessell, Fraser Welsh, Ewan St. John Smith, and David C Bulmer

DOI: 10.1113/JP283170

Corresponding author(s): David Bulmer (dcb53@cam.ac.uk)

The following individual(s) involved in review of this submission have agreed to reveal their identity: Patrick A Hughes (Referee #2); Ken D O'Halloran (Referee #3)

Review Timeline:

Submission Date:	15-Feb-2022
Editorial Decision:	29-Mar-2022
Resubmission Received:	21-May-2022
Editorial Decision:	09-Jun-2022
Revision Received:	26-Jun-2022
Accepted:	28-Jun-2022

Senior Editor: Laura Bennet

Reviewing Editor: Bernard Drumm

Transaction Report:

Dear Dr Bulmer,

Re: JP-RP-2022-282988 "Sensitisation of colonic nociceptors by TNF α is dependent on TNFR1 expression and p38 MAPK activity" by Katie H Barker, James P Higham, Luke A Pattison, Toni Stacey Taylor, Iain P Chessell, Fraser Welsh, Ewan St. John Smith, and David C Bulmer

Thank you for submitting your manuscript to The Journal of Physiology. It has been assessed by a Reviewing Editor and by 2 Referees and the reports are copied below.

Please let your co-authors know of the following editorial decision as quickly as possible.

As you will see, in its current form, the manuscript is not acceptable for publication in The Journal of Physiology. In comments to me, the Reviewing Editor expressed interest in the potential of this study, but much work still needs to be done (and this may include new experiments) in order to satisfactorily address the concerns raised in the reports.

In view of this interest, I would like to offer you the opportunity to carry out all of the changes requested in full, and to resubmit a new manuscript using the "Submit Special Case Resubmission for JP-RP-2022-282988..." on your homepage.

We cannot, of course, guarantee ultimate acceptance at this stage as the revisions required are substantial. However, we encourage you to consider the requested changes and resubmit your work to us if you are able to complete or address all changes.

A new manuscript would be renumbered and redated, but the original referees would be consulted wherever possible. An additional referee's opinion could be sought, if the Reviewing Editor felt it necessary. A full response to each of the reports should be uploaded with a new version.

I hope that the points raised in the reports will be helpful to you.

Yours sincerely,

Professor Kim E. Barrett
Editor-in-Chief
The Journal of Physiology
<https://jp.msubmit.net>
<http://jp.physoc.org>
The Physiological Society
Hodgkin Huxley House
30 Farringdon Lane
London, EC1R 3AW
UK
<http://www.physoc.org>
<http://journals.physoc.org>

EDITOR COMMENTS

Reviewing Editor:

This paper examines the effects of TNF alpha in sensitization of mouse colonic afferent nerves. The authors show that TNF alpha increases Ca²⁺ influx in DRG neurons in response to capsaicin or colonic distention via TNFR1 receptor and TRPV1 channel activation and p38 MAPK signaling. While the study is relatively robust, the referees highlighted concerns regarding the novelty of the study. The role of TRPV1, TNFR1 and p38 in mediating effects of TNF alpha in sensory nerves have been shown in several previous studies, albeit in different models.

The study could be greatly strengthened by providing more novel insight into the mechanism of TNF alpha action in this pathway, several possible means are suggested in the authors discussion but a revised manuscript should explore these possibilities experimentally to yield more mechanistic information on their results. Some points highlighted by the authors include determining if p38 MAPK phosphorylates specific residues of TRPV1 channels in response to stimuli, or how TNF alpha and p38 MAPK signalling affect expression of and activity of TRPA1 or Nav1.8 channels (as only a proportion of the TNF alpha mediated Ca²⁺ influx response was sensitive to TRPV1 blockade).

In addition, the referees have indicated that a revised manuscript will need to include a more detailed and specific methods sections clarifying information on how replicates were obtained and the Ca²⁺ imaging analysis performed, as well revising several figures that require clarification or additional information.

REFEREE COMMENTS

Referee #1:

The study by Barker et al examines the role of TNF α in sensitization to capsaicin and to mechanical stimulation. They used a combination of calcium imaging of DRG neurons and ex vivo colonic afferent nerve recordings. They found that TNF α directly increases calcium influx via the TNFR1 receptor preferentially in capsaicin sensitive DRG neurons. It also sensitizes DRG neurons to capsaicin via TNFR1 and p38 MAPK signaling and attenuates the rundown to capsaicin in these neurons. They then show that TNF α also sensitizes colonic afferent nerves to colonic distension and capsaicin, effects that are also reduced in TNFR1 knockout mice and by blocking p38 MAPK signaling.

The combination of techniques is a nice way to demonstrate the effect of TNF α . While it is likely that most of the DRG neurons recorded from do not innervate the colon, the colonic afferent nerve recordings support the findings in the DRG neurons as occurring the peripheral colonic afferents. Many of the findings in this study confirm that TNF α sensitizes DRG neurons to capsaicin (Nicol et al 1997, Constantin et al 2008); TNF α works via TNFR1 in DRG neurons and colonic afferent nerves (Hughes et al 2013, Ibeakanma and Vanner 2010); and that TNF α can sensitize colonic afferent nerves to mechanical stimulation (Hughes et al 2013). Thus, the major findings are that TNF α reduces the desensitization to capsaicin in DRG neurons and that TNF α sensitizes colonic afferent nerves to capsaicin and colonic distension and this involves p38 MAPK signaling. Consequently, pursuing this relationship/mechanisms would have enhanced the study. Additionally, there are specifics related to the data presented that should be addressed.

1. The study would be strengthened by using more than one capsaicin concentration to demonstrate sensitization by TNF α - is the effect of TNF α seen when lower concentrations of capsaicin are given?
2. Some of the graphs were hard to follow. For example, in Fig5, capsaicin is given in the presence of SB203590+vehicle and SB203590+ TNF α . However, it would be important to have a capsaicin only group to ensure there is no reduction in the capsaicin response by SB203590.
3. In general, references for the concentrations of many of the drugs should be provided as to why that concentration was chosen should be given for many of the drugs used in this study (e.g. A425619, SB203590).
4. It appears that much of the analysis in DRG neurons is done per mouse (i.e. calculate average response of all neurons in each mouse and these results are used in the analysis). In many cases, this means have 5 data points in each group (in some cases only 3 - i.e. Fig4). With smaller numbers, non-parametric tests may need to be employed as opposed to parametric tests that are often used in the analysis in this study. At least when possible one must determine if the data passes normality. If yes, a parametric test could be used; if no, a non-parametric test should be used.
5. Somewhat related, I was a little unclear of the calcium imaging analysis. For example in Fig.1, is it 5 dishes per mouse and 5 mice? Or is it 1 dish per mouse and 5 mice (and thus have 5 dishes total). This could be clarified.
6. I wasn't sure why Fig.2 comes after Fig.1. It was unclear why TNF α 3nM was used in Figure 1 but then that becomes clear that the concentration was likely chosen due to the findings in Fig.1. Thus, could consider reversing the order of these Figures.
7. Fig 2C - this was with 0.1nM TNF α which appears to show much larger responses in a large number of neurons compared to the 0.1nM in Fig. 2A. Similarly, the trace in Fig.2A for 0.03nM shows basically no response yet ~20% of cells respond. I think the amplitude responses to these drug concentrations should be shown. Furthermore, the amplitude of the responses using calcium imaging should be presented in many of the calcium imaging figures as it would show whether in neurons that do respond, whether there is a change in the magnitude of that response.
8. In the photomicrographs in Fig. 4, why does the beta tubulin staining of neurons in the top row look so different than the bottom row? The neurons in the bottom row look much larger than those in the top row. Also, in looking at the overlay in the bottom row, the DAPI staining should fill the area that would be the nucleus but in many cases, it does not. Why is this? Again, showing the amplitude of the responses could further demonstrate a lack of role for glia.
9. Why was TNF α given intraluminal and capsaicin via bath application in the colonic afferent nerve recordings?
10. In Fig 6B, there is significance between Ramp 5 vehicle vs capsaicin. However, isn't the proper comparison Ramp 3 vs 5 vehicle and compare ramp 3 vs 5 TNF α ? If there is a statistical difference between ramp 3 and 5 in the vehicle but no difference between ramp 3 and 5 TNF α , that would indicate that TNF α prevents the run-down to distension. One issue in a multi-unit recordings is that some recordings may have a greater number of units than other recordings and thus have a greater total neural activity. If compare within group rather than between group one could control for this. Or one would have to determine the number of units in a recording and divide the activity by those units to normalize the neural activity across recordings.

Referee #2:

Inflammatory mediators including TNF- α are known to induce pain, however the mechanisms underlying these effects remain largely unknown. Here the authors build on previous studies using a combination of Ca imaging and electrophysiology in w/t and TNFR1 k/o mice to demonstrate the importance of TNF- α in neuronal bodies and afferent endings in the gut and delineate the role of TRPV1, p38 MAPK and PKC signalling pathways. TNF- α has previously been shown to sensitise or activate nerves via TNFR1, TRPV1 and p38MAPK in several combined manuscripts, however these are largely disjointed and the novelty here is the thorough examination of the TNF signalling pathway in a single organ. The manuscript is well written with a clear aim, robust methods and conclusions supported by the results. There are some concerns:

Fig 1F - Prevention of capsaicin desensitisation by TNF- α . Further clarity is required for the description of capsaicin responsive / non responsive neurons and the proportion of capsaicin responsive. The text may read that 37.7 \pm 14.2% of neurons were capsaicin responsive?? What was the proportion of capsaicin non-responsive neurons, and capsaicin responsive but TNF- α non-responsive neurons.

Fig 2: The text reads 'application of TNF- α elicited a concentration dependent....., and the proportion of neurons activated by TNF- α '. The second part of this statement requires clarity. How does application of TNF- α increase the proportion of neurons activated by TNF- α ? Were there two TNF- α applications?

Fig 4B: How were neurons / non-neurons counted? This detail is lacking in the methods.

Discussion: '....ability of capsaicin to evoked significant pain.....' Evoke, not evoked.

ADDITIONAL FORMATTING REQUIREMENTS:

- Author photo and profile. First (or joint first) authors are asked to provide a short biography (no more than 100 words for one author or 150 words in total for joint first authors) and a portrait photograph. These should be uploaded and clearly labelled with the revised version of the manuscript. See Information for Authors for further details.
- Your manuscript must include a complete Additional Information section
- Please upload separate high-quality figure files via the submission form.
- Please ensure that the Article File you upload is a Word file.
- A Statistical Summary Document, summarising the statistics presented in the manuscript, is required upon revision. It must be on the Journal's template, which can be downloaded from the link in the Statistical Summary Document section here: https://jp.msubmit.net/cgi-bin/main.plex?form_type=display_requirements#statistics
- Papers must comply with the Statistics Policy https://jp.msubmit.net/cgi-bin/main.plex?form_type=display_requirements#statistics

In summary:

- If n {less than or equal to} 30, all data points must be plotted in the figure in a way that reveals their range and distribution. A bar graph with data points overlaid, a box and whisker plot or a violin plot (preferably with data points included) are acceptable formats.
- If $n > 30$, then the entire raw dataset must be made available either as supporting information, or hosted on a not-for-profit repository e.g. FigShare, with access details provided in the manuscript.
- 'n' clearly defined (e.g. x cells from y slices in z animals) in the Methods. Authors should be mindful of pseudoreplication.
- All relevant 'n' values must be clearly stated in the main text, figures and tables, and the Statistical Summary Document (required upon revision).
- The most appropriate summary statistic (e.g. mean or median and standard deviation) must be used. Standard Error of the Mean (SEM) alone is not permitted.
- Exact p values must be stated. Authors must not use 'greater than' or 'less than'. Exact p values must be stated to three significant figures even when 'no statistical significance' is claimed.
- Statistics Summary Document completed appropriately upon revision.
- Please include an Abstract Figure. The Abstract Figure is a piece of artwork designed to give readers an immediate understanding of the research and should summarise the main conclusions. If possible, the image should be easily

'readable' from left to right or top to bottom. It should show the physiological relevance of the manuscript so readers can assess the importance and content of its findings. Abstract Figures should not merely recapitulate other figures in the manuscript. Please try to keep the diagram as simple as possible and without superfluous information that may distract from the main conclusion(s). Abstract Figures must be provided by authors no later than the revised manuscript stage and should be uploaded as a separate file during online submission labelled as File Type 'Abstract Figure'. Please ensure that you include the figure legend in the main article file. All Abstract Figures should be created using BioRender. Authors should use The Journal's premium BioRender account to export high-resolution images. Details on how to use and access the premium account are included as part of this email.

Confidential Review

15-Feb-2022

Rebuttal Letter

We would like to thank the Editor and Reviewers for their thoughtful and insightful comments which have greatly improved our manuscript. We have completed additional experiments and amended the manuscript to address these concerns. Please find our point-by-point response to issues flagged by the Editor and Reviewers following the initial submission of this paper.

Reviewing Editor:

The study could be greatly strengthened by providing more novel insight into the mechanism of TNF alpha action in this pathway, several possible means are suggested in the authors discussion but a revised manuscript should explore these possibilities experimentally to yield more mechanistic information on their results. Some points highlighted by the authors include determining if p38 MAPK phosphorylates specific residues of TRPV1 channels in response to stimuli, or how TNF alpha and p38 MAPK signalling affect expression of and activity of TRPA1 or Nav1.8 channels (as only a proportion of the TNF alpha mediated Ca²⁺ influx response was sensitive to TRPV1 blockade).

We thank the Editor and agree that more insight into the mechanism of TNF α -mediated neuronal stimulation was needed. In response to these recommendations, we have performed additional experiments looking at TRPA1 involvement in TNF α -mediated increases in [Ca²⁺]_i, consistent with the narrative of modulated TRP channel signalling to TNF α . We have included these data showing a contribution of both TRPV1 and TRPA1 to TNF α -mediated Ca²⁺ influx. The combination of TRPV1 and TRPA1 blockade abolished Ca²⁺ responses in DRG neurons, consistent with published data showing a role for TRPA1 in TNF α -mediated afferent sensitisation and collectively these data greatly strengthen our understanding of TNF α -mediated signalling. Regarding determination of which TRPV1 residues undergo p38 MAPK phosphorylation, whilst we agree that these studies would be informative, it would require a significant number of animals to harvest sufficient neurones to measure. Moreover, generation of a knock-in mouse with the proposed TRPV1 site mutated would be required to provide a definitive answer. Although experiments could be conducted in cell lines, this would require establishment of a line with stable expression of both TRPV1 and TNFR1 (and subsequent mutant TRPV1(s)) and would still need validating in DRG neurons. Thus, we believe that such an investigation should be the subject of a follow-on project and that our current speculation of p38 MAPK phosphorylation sites on TRPV1 is sufficient.

In addition, the referees have indicated that a revised manuscript will need to include a more detailed and specific methods sections clarifying information on how replicates were obtained and the Ca²⁺ imaging analysis performed, as well revising several figures that require clarification or additional information.

These changes have now been made and are addressed in detail below.

Referee #1

1. The study would be strengthened by using more than one capsaicin concentration to demonstrate sensitization by TNF α - is the effect of TNF α seen when lower concentrations of capsaicin are given?

We agree with the Reviewer's comments and subsequently assessed lower concentrations of capsaicin. Lower concentrations of capsaicin produced a submaximal, concentration dependent increase in intracellular [Ca²⁺]_i and proportion of neurons activated. There was a trend towards the sensitisation of these responses following TNF α administration but these differences were not significant with two-way ANOVA. Data shown in Figure 1.

2. Some of the graphs were hard to follow. For example, in Fig5, capsaicin is given in the presence of SB203590+vehicle and SB203590+ TNF α . However, it would be important to have a capsaicin only group to ensure there is no reduction in the capsaicin response by SB203590.

To clarify the potential impact of SB203580 treatment alone, we have included the response to capsaicin alone in Figure 5. No difference is observed in the magnitude of capsaicin responses or proportion of capsaicin-sensitive neurons in the presence vs. absence of SB203580. The figure legend has been updated to clarify that the capsaicin alone data set is also presented in Figure 1. Statistical tests have been updated accordingly.

3. In general, references for the concentrations of many of the drugs should be provided as to why that concentration was chosen should be given for many of the drugs used in this study (e.g. A425619, SB203590).

We thank the Reviewer for identifying this point. References have been inserted highlighting previous studies from which concentrations were determined for use in this study.

4. It appears that much of the analysis in DRG neurons is done per mouse (i.e. calculate average response of all neurons in each mouse and these results are used in the analysis). In many cases, this means have 5 data points in each group (in some cases only 3 - i.e. Fig4). With smaller numbers, non-parametric tests may need to be employed as opposed to parametric tests that are often used in the analysis in this study. At least when possible one must determine if the data passes normality. If yes, a parametric test could be used; if no, a non-parametric test should be used.

Thank you for raising this point. We reviewed the distribution of data sets and made adjustments to the use of non-parametric tests where appropriate. Further information has been included in the data analysis and figure legends as required; a highlighted version of the manuscript has been submitted to clarify these and other changes.

5. Somewhat related, I was a little unclear of the calcium imaging analysis. For example in Fig.1, is it 5 dishes per mouse and 5 mice? Or is it 1 dish per mouse and 5 mice (and thus have 5 dishes total). This could be clarified.

Thank you for highlighting the need for clarification. For the Ca²⁺ imaging analysis, n refers to the total number of dishes and N refers to the total number of mice i.e. n = 5, N = 5 represents 1 dish per mouse from 5 mice (5 dishes in total). In figures, each data point represents an individual dish. Methods have been updated to make this clearer.

6. I wasn't sure why Fig.2 comes after Fig.1. It was unclear why TNF α 3nM was used in Figure 1 but then that becomes clear that the concentration was likely chosen due to the findings in Fig.1. Thus, could consider reversing the order of these Figures.

Our initial studies shown in Figure 1 were carried out with 3 nM TNF α , consistent with previous studies (Hughes et al., 2013). Following the observation of a TNF α -mediated increase in [Ca²⁺]_i in DRG neurons, we went on to generate a concentration response curve to further characterise the response, hence the sequence of these figures.

7. Fig 2C - this was with 0.1nM TNF α which appears to show much larger responses in a large number of neurons compared to the 0.1nM in Fig. 2A. Similarly, the trace in Fig.2A for 0.03nM shows basically no response yet ~20% of cells respond. I think the amplitude responses to these drug concentrations should be shown. Furthermore, the amplitude of the responses using calcium

imaging should be presented in many of the calcium imaging figures as it would show whether in neurons that do respond, whether there is a change in the magnitude of that response.

We agree that responses at lower concentrations are lower in magnitude and have kept the same scaling for each of the concentrations to highlight this difference. We appreciate this gives the impression that not many cells are responding at lower concentrations, however, after applying our 10% cut off, there were still a reasonable number of cells demonstrating smaller Ca^{2+} responses.

We agree that the addition of response amplitude would greatly improve the manuscript and have therefore now included these data for all Ca^{2+} imaging analysis and updated the text accordingly.

8. In the photomicrographs in Fig. 4, why does the beta tubulin staining of neurons in the top row look so different than the bottom row? The neurons in the bottom row look much larger than those in the top row. Also, in looking at the overlay in the bottom row, the DAPI staining should fill the area that would be the nucleus but in many cases, it does not. Why is this? Again, showing the amplitude of the responses could further demonstrate a lack of role for glia.

We thank the Reviewer for highlighting this inconsistency. In MACS cultures (top row), some larger diameter neurons are retained in the column and cultured neurons therefore tend to be smaller in appearance compared to unsorted (bottom row). DAPI staining in the larger neurons (bottom row) is dependent on the location of the nucleus which may not be in the centre of the cell, or occupy the majority of the cell. If the nucleus is located close to the membrane, the signal may be more difficult to see due to β III-tubulin staining. No change in the magnitude of $\text{TNF}\alpha$ responses was seen in MACS compared to unsorted cultures, further confirming glial cells are not required for $\text{TNF}\alpha$ -mediated activation. These data has now been added to Figure 4.

9. Why was $\text{TNF}\alpha$ given intraluminal and capsaicin via bath application in the colonic afferent nerve recordings?

Bath application requires a much larger volume of drug to be added making the delivery of $\text{TNF}\alpha$ via this route prohibitively expensive at the concentrations tested. Intraluminal administration of drug was therefore used as an alternative.

10. In Fig 6B, there is significance between Ramp 5 vehicle vs capsaicin. However, isn't the proper comparison Ramp 3 vs 5 vehicle and compare ramp 3 vs 5 $\text{TNF}\alpha$? If there is a statistical difference between ramp 3 and 5 in the vehicle but no difference between ramp 3 and 5 $\text{TNF}\alpha$, that would indicate that $\text{TNF}\alpha$ prevents the run-down to distension. One issue in a multi-unit recordings is that some recordings may have a greater number of units than other recordings and thus have a greater total neural activity. If compare within group rather than between group one could control for this. Or one would have to determine the number of units in a recording and divide the activity by those units to normalize the neural activity across recordings.

We thank the Reviewer for highlighting this point and agree interpretation of these data is complex because both a trend towards desensitisation in ramp responses occurs over time and an increase in response is seen with $\text{TNF}\alpha$. We feel that the best way to present these data, accounting for the different changes in ramp distension responses over time, is by time matched comparisons of vehicle and treated groups. We do agree that comparisons (of ramp 3 and ramp 5) within experiments are also important and have included a description of this data and statistical analysis within the written results of the manuscript. Comparisons within treatment groups demonstrated a significant increase in afferent response between ramp 3 and 5 in $\text{TNF}\alpha$ -treated tissues, and no significant change between ramp 3 and ramp 5 responses in vehicle treated tissues.

We agree that differences in the number of fibres recorded may exist between experiments, but, since the AUC of Ramp 3 was no different between vehicle and TNF α -treated groups, we believe there is sufficient parity between studies to permit the direct comparison of ramp 5.

Referee #2

1. Fig 1F - Prevention of capsaicin desensitisation by TNF-a. Further clarity is required for the description of capsaicin responsive / non responsive neurons and the proportion of capsaicin responsive. The text may read that 37.7+/- 14.2% of neurons were capsaicin responsive?? What was the proportion of capsaicin non-responsive neurons, and capsaicin responsive but TNF-a non-responsive neurons.

We agree further clarification is needed. The text now reads "This effect was only observed in a subset of capsaicin-responsive neurons (29.93 \pm 11.76%) that were co-sensitive to TNF α and represented 82.15 \pm 20.91% of total capsaicin responders"

2. Fig 2: The text reads 'application of TNF-a elicited a concentration dependent....., and the proportion of neurons activated by TNF-a'. The second part of this statement requires clarity. How does application of TNF-a increase the proportion of neurons activated by TNF-a? Were there two TNF-a applications?

We acknowledge that this was not clear. The number of neurons reaching the responder threshold increases when higher concentrations of TNF α are applied – this has now been clarified in the text.

3. Fig 4B: How were neurons / non-neurons counted? This detail is lacking in the methods.

As described in the Image Analysis section of the methods, following an automatic threshold algorithm, determined by ImageJ, positively stained particles were automatically counted to calculate the ratio of β III-tubulin-positive cells (neurons) to DAPI-positive cells (neurons and satellite cells) – represented as a % in Figure 4B.

4. Discussion: '....ability of capsaicin to evoked significant pain.....' Evoke, not evoked.

Now reads "ability of capsaicin to evoke significant pain".

Dear Dr Bulmer,

Re: JP-RP-2022-283170X "Sensitisation of colonic nociceptors by TNF α is dependent on TNFR1 expression and p38 MAPK activity" by Katie H Barker, James P Higham, Luke A Pattison, Toni Stacey Taylor, Iain P Chessell, Fraser Welsh, Ewan St. John Smith, and David C Bulmer

Thank you for submitting your manuscript to The Journal of Physiology. It has been assessed by a Reviewing Editor and by 3 expert Referees and I am pleased to tell you that it is considered to be acceptable for publication following satisfactory revision.

The reports are copied at the end of this email. Please address all of the points and incorporate all requested revisions, or explain in your Response to Referees why a change has not been made.

NEW POLICY: In order to improve the transparency of its peer review process The Journal of Physiology publishes online as supporting information the peer review history of all articles accepted for publication. Readers will have access to decision letters, including all Editors' comments and referee reports, for each version of the manuscript and any author responses to peer review comments. Referees can decide whether or not they wish to be named on the peer review history document.

Authors are asked to use The Journal's premium BioRender (<https://biorender.com/>) account to create/redraw their Abstract Figures. Information on how to access The Journal's premium BioRender account is here:

<https://physoc.onlinelibrary.wiley.com/journal/14697793/biorender-access> and authors are expected to use this service. This will enable Authors to download high-resolution versions of their figures. The link provided should only be used for the purposes of this submission. Authors will be charged for figures created on this premium BioRender account if they are not related to this manuscript submission.

I hope you will find the comments helpful and have no difficulty returning your revisions within 4 weeks.

Your revised manuscript should be submitted online using the links in Author Tasks Link Not Available.

Any image files uploaded with the previous version are retained on the system. Please ensure you replace or remove all files that have been revised.

REVISION CHECKLIST:

- Article file, including any tables and figure legends, must be in an editable format (eg Word)
- Abstract figure file (see above)
- Statistical Summary Document
- Upload each figure as a separate high quality file
- Upload a full Response to Referees, including a response to any Senior and Reviewing Editor Comments;
- Upload a copy of the manuscript with the changes highlighted.

- A potential 'Cover Art' file for consideration as the Issue's cover image;
- Appropriate Supporting Information (Video, audio or data set https://jp.msubmit.net/cgi-bin/main.plex?form_type=display_requirements#supp).

To create your 'Response to Referees' copy all the reports, including any comments from the Senior and Reviewing Editors, into a Word, or similar, file and respond to each point in colour or CAPITALS and upload this when you submit your revision.

I look forward to receiving your revised submission.

If you have any queries please reply to this email and staff will be happy to assist.

Yours sincerely,

Professor Laura Bennet
Senior Editor
The Journal of Physiology
<https://jp.msubmit.net>
<http://jp.physoc.org>
The Physiological Society
Hodgkin Huxley House
30 Farringdon Lane
London, EC1R 3AW
UK
<http://www.physoc.org>
<http://journals.physoc.org>

REQUIRED ITEMS:

- Author photo and profile. First (or joint first) authors are asked to provide a short biography (no more than 100 words for one author or 150 words in total for joint first authors) and a portrait photograph. These should be uploaded and clearly labelled with the revised version of the manuscript. See Information for Authors for further details.
- The contact information provided for the person responsible for 'Research Governance' at your institution is an author on this paper. Please provide an alternative contact who is not an author on this paper or confirm that the author whose email was provided has sole responsibility for research governance. This is the person who is responsible for regulations, principles and standards of good practice in research carried out at the institution, for instance the ethical treatment of animals, the keeping of proper experimental records or the reporting of results.
- Your manuscript must include a complete Additional Information section.
- The Journal of Physiology funds authors of provisionally accepted papers to use the premium BioRender site to create high resolution schematic figures. Follow this link and enter your details and the manuscript number to create and download figures. Upload these as the figure files for your revised submission. If you choose not to take up this offer we require figures to be of similar quality and resolution. If you are opting out of this service to authors, state this in the Comments section on the Detailed Information page of the submission form. The link provided should only be used for the purposes of this submission. Authors will be charged for figures created on this premium BioRender account if they are not related to this manuscript submission.
- Papers must comply with the Statistics Policy: https://jp.msubmit.net/cgi-bin/main.plex?form_type=display_requirements#statistics.

In summary:

- If $n \leq 30$, all data points must be plotted in the figure in a way that reveals their range and distribution. A bar graph with data points overlaid, a box and whisker plot or a violin plot (preferably with data points included) are acceptable formats.
- If $n > 30$, then the entire raw dataset must be made available either as supporting information, or hosted on a not-for-profit repository e.g. FigShare, with access details provided in the manuscript.
- 'n' clearly defined (e.g. x cells from y slices in z animals) in the Methods. Authors should be mindful of pseudoreplication.
- All relevant 'n' values must be clearly stated in the main text, figures and tables, and the Statistical Summary Document (required upon revision).
- The most appropriate summary statistic (e.g. mean or median and standard deviation) must be used. Standard Error of the Mean (SEM) alone is not permitted.
- Exact p values must be stated. Authors must not use 'greater than' or 'less than'. Exact p values must be stated to three significant figures even when 'no statistical significance' is claimed.
- Statistics Summary Document completed appropriately upon revision
- Please include an Abstract Figure. The Abstract Figure is a piece of artwork designed to give readers an immediate understanding of the research and should summarise the main conclusions. If possible, the image should be easily 'readable' from left to right or top to bottom. It should show the physiological relevance of the manuscript so readers can

assess the importance and content of its findings. Abstract Figures should not merely recapitulate other figures in the manuscript. Please try to keep the diagram as simple as possible and without superfluous information that may distract from the main conclusion(s). Abstract Figures must be provided by authors no later than the revised manuscript stage and should be uploaded as a separate file during online submission labelled as File Type 'Abstract Figure'. Please ensure that you include the figure legend in the main article file. All Abstract Figures should be created using BioRender. Authors should use The Journal's premium BioRender account to export high-resolution images. Details on how to use and access the premium account are included as part of this email.

EDITOR COMMENTS

Reviewing Editor:

The authors have revised their manuscript and provided more mechanistic insight into the sensitization pathway that they propose. There are still some queries raised by reviewer 1 that remain to be addressed. The authors should consider the proposed experiment examining Ca²⁺ responses in DRG neurons evoked by a TRPA1 agonist and how these may be affected by the p38 MAPK pathway. This would be a valuable mechanistic addition to the study and would tie in more with the proposed schematic in the abstract summary figure.

The authors should also expand their key points summary as it is currently only 3 short bullet points. Addition of an additional 1-2 points highlighting the context and potential impact of the work would be welcome.

REFEREE COMMENTS

Referee #1:

This is a resubmission by Barker et al examining the role of TNF α in sensitization to capsaicin and to mechanical stimulation. The authors have completed some new experiments including using lower concentrations of capsaicin to see if sensitization occurred at these lower concentrations, and demonstrating a role for TRPA1 in TNF α induced increase in intracellular calcium by DRG neurons, and did some further analysis. These revisions add some important data to this study. However, I think there are some further issues that need to be addressed.

1. Why did they authors only examine the role of TRPA1 on DRG neuron calcium responses evoked by TNF α ? Why not examine if the same TNF α pathway involving p38 MAPK for TRPV1 is also involved in sensitization of TRPA1 using a TRPA1 agonist on DRG neurons or colonic afferent nerves? This could determine if there is sensitization of multiple TRP channels via this pathway.
2. The authors clarified that it was often 1 dish per mouse for calcium imaging experiments. I may have missed it, but what was the range of neurons per dish? That would then give an idea of how many neurons were recorded in each dataset.
3. I think there needs to be some clarification in the manuscript regarding Fig.4A. If the technique involves sorting cells using a non-neuronal antibody cocktail why are large neuronal cells being left in the column? If this is the case with this technique, that should be stated and a reference provided in the manuscript as the 2 middle panels look very different and would be confusing without that explanation. It is good that the authors performed functional experiments after the sorting technique demonstrating that after sorting the cells are healthy.
4. I think the wording that TNF α prevents desensitization to ramp distension needs to be revised. Some recordings will show a decrease in the response to distension over repeated distensions (an example is the recording shown in the Fig 6) but other recordings will remain stable. When looking at the data in Fig 6, the means are similar and I gather there is no significance between ramp 3 and 5. The authors rebuttal also states that there is a trend to desensitization rather than then there being a desensitization. Thus, I don't think the wording should be it reduces the desensitization; perhaps more simply could state that TNF α sensitizes the nerves to mechanical distension.

Referee #2:

The authors have improved their manuscript by performing additional experiments and altering text in line with the reviewers comments. I have no further comments.

Referee #3:

Thank you for submitting your manuscript to The Journal of Physiology. Details of ethical approval are provided by the authors. The source and sex of the animals used is clearly stated. The method of killing is clearly stated and is an approved method (rising concentration of CO₂ followed by cervical dislocation).

END OF COMMENTS

1st Confidential Review

21-May-2022

Rebuttal Letter

We would once again like to thank the editor and reviewers for their thoughtful and insightful comments which have improved our manuscript. Amendments have been made to address their concerns. Please find our response to each point raised below.

Reviewing Editor:

The authors have revised their manuscript and provided more mechanistic insight into the sensitization pathway that they propose. There are still some queries raised by reviewer 1 that remain to be addressed. The authors should consider the proposed experiment examining Ca²⁺ responses in DRG neurons evoked by a TRPA1 agonist and how these may be affected by the p38 MAPK pathway. This would be a valuable mechanistic addition to the study and would tie in more with the proposed schematic in the abstract summary figure.

The points raised are addressed in detail below. The graphical abstract has been amended to reflect more clearly the findings of the paper.

The authors should also expand their key points summary as it is currently only 3 short bullet points. Addition of an additional 1-2 points highlighting the context and potential impact of the work would be welcome.

Further bullet points have now been added to key points summary.

REFeree COMMENTS

Referee #1:

This is a resubmission by Barker et al examining the role of TNF α in sensitization to capsaicin and to mechanical stimulation. The authors have completed some new experiments including using lower concentrations of capsaicin to see if sensitization occurred at these lower concentrations, and demonstrating a role for TRPA1 in TNF α induced increase in intracellular calcium by DRG neurons, and did some further analysis. These revisions add some important data to this study. However, I think there are some further issues that need to be addressed.

1. Why did they authors only examine the role of TRPA1 on DRG neuron calcium responses evoked by TNF α ? Why not examine if the same TNF α pathway involving p38 MAPK for TRPV1 is also involved in sensitization of TRPA1 using a TRPA1 agonist on DRG neurons or colonic afferent nerves? This could determine if there is sensitization of multiple TRP channels via this pathway.

We agree with the reviewer that these experiments are now the natural follow up study to our additional work revealing a contribution by TRPA1 to TNF mediated Ca²⁺ flux in DRG neurons and build on the studies by Hughes et al 2013, showing a role for TRPA1 in TNF augmented colonic afferent mechanosensitivity. This was one of the original driving forces behind this paper, and a basis of our initial decision to focus on capsaicin/TRPV1 signalling which had remained unstudied for colonic afferents.

Unfortunately, we are not able to prosecute the experiments described in the timeframe prescribed. The TNFR1-/- mice are currently breeding poorly (for example, unexpected deaths of pups and a breeding female have lead Condition 18 reports being submitted to the Home Office in recent months) and will take some time for sufficient mice to be available at the correct age range

We respectfully believe these experiments are best suited to our follow up to study in which we will investigate the role for p38MAPK in the sensitisation of multiple TRP channels, including TRPA1, TRPV4 and TRPM8 by TNF alpha.

2. The authors clarified that it was often 1 dish per mouse for calcium imaging experiments. I may have missed it, but what was the range of neurons per dish? That would then give an idea of how many neurons were recorded in each dataset.

Average number of neurons per dish has now been added.

3. I think there needs to be some clarification in the manuscript regarding Fig.4A. If the technique involves sorting cells using a non-neuronal antibody cocktail why are large neuronal cells being left in the column? If this is the case with this technique, that should be stated and a reference provided in the manuscript as the 2 middle panels look very different and would be confusing without that explanation. It is good that the authors performed functional experiments after the sorting technique demonstrating that after sorting the cells are healthy.

This has been added into the text.

4. I think the wording that TNF α prevents desensitization to ramp distension needs to be revised. Some recordings will show a decrease in the response to distension over repeated distensions (an example is the recording shown in the Fig 6) but other recordings will remain stable. When looking at the data in Fig 6, the means are similar and I gather there is no significance between ramp 3 and 5. The authors rebuttal also states that there is a trend to desensitization rather than then there being a desensitization. Thus, I don't think the wording should be it reduces the desensitization; perhaps more simply could state that TNF α sensitizes the nerves to mechanical distension.

The manuscript has been altered accordingly.

Referee #2:

The authors have improved their manuscript by performing additional experiments and altering text in line with the reviewers comments. I have no further comments.

Referee #3:

Thank you for submitting your manuscript to The Journal of Physiology. Details of ethical approval are provided by the authors. The source and sex of the animals used is clearly stated. The method of killing is clearly stated and is an approved method (rising concentration of CO₂ followed by cervical dislocation).

Dear Dr Bulmer,

Re: JP-RP-2022-283170XR1 "Sensitisation of colonic nociceptors by TNF α is dependent on TNFR1 expression and p38 MAPK activity" by Katie H Barker, James P Higham, Luke A Pattison, Toni Stacey Taylor, Iain P Chessell, Fraser Welsh, Ewan St. John Smith, and David C Bulmer

I am pleased to tell you that your paper has been accepted for publication in The Journal of Physiology.

NEW POLICY: In order to improve the transparency of its peer review process The Journal of Physiology publishes online as supporting information the peer review history of all articles accepted for publication. Readers will have access to decision letters, including all Editors' comments and referee reports, for each version of the manuscript and any author responses to peer review comments. Referees can decide whether or not they wish to be named on the peer review history document.

The last Word version of the paper submitted will be used by the Production Editors to prepare your proof. When this is ready you will receive an email containing a link to Wiley's Online Proofing System. The proof should be checked and corrected as quickly as possible.

Authors should note that it is too late at this point to offer corrections prior to proofing. The accepted version will be published online, ahead of the copy edited and typeset version being made available. Major corrections at proof stage, such as changes to figures, will be referred to the Reviewing Editor for approval before they can be incorporated. Only minor changes, such as to style and consistency, should be made a proof stage. Changes that need to be made after proof stage will usually require a formal correction notice.

All queries at proof stage should be sent to TJP@wiley.com.

Are you on Twitter? Once your paper is online, why not share your achievement with your followers. Please tag The Journal (@jphysiol) in any tweets and we will share your accepted paper with our 23,000+ followers!

Yours sincerely,

Professor Laura Bennet
Senior Editor
The Journal of Physiology
<https://jp.msubmit.net>
<http://jp.physoc.org>
The Physiological Society
Hodgkin Huxley House
30 Farringdon Lane
London, EC1R 3AW
UK
<http://www.physoc.org>
<http://journals.physoc.org>

P.S. - You can help your research get the attention it deserves! Check out Wiley's free Promotion Guide for best-practice recommendations for promoting your work at www.wileyauthors.com/eeo/guide. And learn more about Wiley Editing Services which offers professional video, design, and writing services to create shareable video abstracts, infographics, conference posters, lay summaries, and research news stories for your research at www.wileyauthors.com/eeo/promotion.

*** IMPORTANT NOTICE ABOUT OPEN ACCESS ***

To assist authors whose funding agencies mandate public access to published research findings sooner than 12 months after publication The Journal of Physiology allows authors to pay an open access (OA) fee to have their papers made freely available immediately on publication.

You will receive an email from Wiley with details on how to register or log-in to Wiley Authors Services where you will be able to place an OnlineOpen order.

You can check if your funder or institution has a Wiley Open Access Account here <https://authorservices.wiley.com/author-resources/Journal-Authors/licensing-and-open-access/open-access/author-compliance-tool.html>

Your article will be made Open Access upon publication, or as soon as payment is received.

If you wish to put your paper on an OA website such as PMC or UKPMC or your institutional repository within 12 months of publication you must pay the open access fee, which covers the cost of publication.

OnlineOpen articles are deposited in PubMed Central (PMC) and PMC mirror sites. Authors of OnlineOpen articles are permitted to post the final, published PDF of their article on a website, institutional repository, or other free public server, immediately on publication.

Note to NIH-funded authors: The Journal of Physiology is published on PMC 12 months after publication, NIH-funded authors DO NOT NEED to pay to publish and DO NOT NEED to post their accepted papers on PMC.

EDITOR COMMENTS

Reviewing Editor:

Thank you for revising your manuscript. The reviewers have no further queries.

REFEREE COMMENTS

Referee #1:

Looks good. I have no further comments.

2nd Confidential Review

26-Jun-2022